# Smart event-triggered MINFLUX microscopy to catch and follow rare events

Jonatan Alvelid [1,2,3] ✉, Agnes Koerfer[1,3] & Christian Eggeling [1,3,4] ✉

MINFLUX microscopy allows characterization of molecular organization and dynamics with single nanometer spatial resolution and sub-hundred microseconds temporal resolution. However, acquisition times often span minutes to hours as a single fluorophore is measured at a time. Studying live cellular processes therefore requires careful consideration of where and when to apply it, hence manual control limits its potential applications. To overcome acquisition speed, initiation, and data throughput limitations, we present event-triggered MINFLUX: a smart microscopy method using confocal monitoring with real-time image analysis, and applying MINFLUX exactly where and when deemed necessary. The method is controlled through a custom-written open-source Python framework automatically controlling a commercial MINFLUX microscope. We investigate molecular membrane dynamics and organization in 2D and 3D during cellular events: lipid dynamics at caveolae; membrane topography during dynamin-mediated endocytosis; and membrane fluidity and topography during HIV-1 budding site formation. Rapid event detection and minimal regions of interest provides data that would be unfeasible or impossible to acquire through manual microscope control.

MINFLUX (minimal fluorescence photon fluxes) microscopy[1–4] excels in spatial resolution and temporal tracking resolution, where it reaches single nanometers and below 100 microseconds also in commercial instrumention[5]. The power comes at the cost of lengthy recording times of minutes to hours, as only a single fluorophore is tracked at a time, severely limiting tracking data throughput as compared to conventional single-particle tracking approaches. Applications have thus so far ranged from imaging bacterial[6] or synaptic[7] proteins in fixed cells, to tracking individual motor proteins[8–10] or cargo transport through the nuclear pore[11]. MINFLUX has only sparsely been applied in living cells to capture short-lived seconds-scale events or tracking faster molecules during events of interest, and not for the likes of endocytosis and tracking lipids in cellular membranes.

Smart microscopy[12,13] and sample-adaptive acquisition approaches has helped conventional and super-resolution microscopy methods to improve light exposure, recording times, image quality, and deep-tissue imaging[14–20]. Multiscale and multimodal approaches have been applied to improve conventional microscopy[21], STED (stimulated emission depletion) microscopy[22,23], SMLM (single-molecule localization microscopy)[24], light-sheet microscopy[25,26] and SIM (structured illumination microscopy)[27], and have allowed imaging of both large-scale and small-scale events, slow and fast, from embryo development[28], through cellular division[29–31], to synaptic vesicle recycling[22]. Event-triggered methods[14,21,22,27] are of particular interest, thanks to their ability to spot fleeting biological events otherwise difficult to capture with their respective microscopy methods. In these cases, an event is defined as any output observable using a monitoring microscopy method and labeling of choice, usually dynamic subcellular changes of any kind, such as protein accumulation, shape changes, or contact sites, with a fast, low-resolution, and low-impact method[22]. The concept is also applicable to static structures, and is then effectively ran on a single monitoring image and used as a region-of-interest finder. As an important aspect, open source software has in many cases been released to allow widespread adaptation of the

[1]Institute for Applied Optics and Biophysics, Friedrich Schiller University Jena, Jena, Germany. [2]Department of Applied Physics and SciLifeLab, KTH Royal Institute of Technology, Stockholm, Sweden. [3]Leibniz Institute of Photonic Technology, Jena, Germany. [4]Jena Center for Soft Matter, Jena, Germany. ✉e-mail: jonatan.alvelid@scilifelab.se; christian.eggeling@uni-jena.de

techniques[22,24,29,31,32]. Hitherto, no smart microscopy methods have been applied to MINFLUX, where the potential benefits of increasing the throughput and speeding up acquisitions by limiting regions of interest (ROIs) temporally and spatially are vast[4]. Such a method could enable MINFLUX to achieve a higher throughput of repetitive cellular events, and access rare cellular events on the second timescale, providing 2D and 3D data of unrivaled quality.

Here, we present event-triggered MINFLUX (etMINFLUX) microscopy, where MINFLUX acquisitions (2D or 3D, imaging or tracking) at the site and time of an event of interest are triggered by the outcome of real-time analysis of confocal measurements. Confocal imaging up to 1 Hz is used to monitor the sample for an event of interest, and when captured, the microscope switches to a MINFLUX acquisition. We develop a Python-based-framework implementation of the method and apply it to three different biological systems and cellular events, proving the versatility of etMINFLUX. Firstly, the throughput of lipid tracking at caveolae sites is highly increased, using the intensity peak of static and fluorescently labeled Caveolin1 agglomerations as the event; secondly, endosomes budding from the membrane are detected, using appearance, accumulation, and consistent stationary presence of fluorescently labeled Dynamin1 as the event, and mapped in 3D on a timescale of tens of seconds; and thirdly, Gag-protein accumulation sites as an indicator of HIV-1 (human immunodeficiency virus) budding sites are detected, using appearance, accumulation, consistent stationary presence, and isolated position of fluorescently labeled Gag as the event, and the membrane fluidity and shape is followed over a timescale of minutes.

## Results

### Event-triggered MINFLUX method and practical framework

In order to alleviate the limitations in MINFLUX of long recording times, and allowing the capabilities of the method to be applied to investigate more dynamic cellular processes otherwise inaccessible, we developed etMINFLUX. This smart microscopy method can trigger MINFLUX tracking or imaging immediately upon detection of dynamic cellular events in confocal images (Fig. 1a). The method uses relatively low-resolution confocal microscopy to monitor the sample, which can be readily implemented on any MINFLUX microscope due to their shared key optical components. Confocal imaging enables monitoring of large sample regions, and real-time analysis pipelines applied to the

resulting images can detect predefined events of interest on millisecond timescales, which is much faster than manual control, forming the first key concept of the method. Upon event detection, MINFLUX microscopy is rapidly applied in the immediate vicinity of the event site, typically within hundreds of milliseconds (Fig. 1b). As a second key concept and advantage, the MINFLUX ROIs are sufficiently small to allow rapid sampling of the entire ROI, typically on the order of seconds when tracking diffusing molecules, thereby enabling the collection of a substantial number of relevant molecular trajectories. While the overall data throughput remains intrinsically limited by the sequential single-molecule-tracking nature of MINFLUX, etMINFLUX maximizes the fraction of recorded trajectories that are informative. Finally, as a third key concept and advantage, the approach limits the excitation exposure during MINFLUX acquisitions to the precise locations and times where information is required. Although confocal imaging is used to monitor the sample, similar confocal illumination is also necessary for manual MINFLUX ROI selection. Combined with the increased temporal efficiency of acquiring relevant MINFLUX data and comparable effective illumination intensities between confocal and MINFLUX acquisition[3,4], etMINFLUX reduces overall light exposure and thereby helps minimize phototoxicity.

We implemented the method in a practical framework directly applicable to a commercial abberior MINFLUX microscope. In this implementation, sample regions up to $80 \times 80 \, \mu m^2$ can be monitored with confocal imaging, and the resulting images are analyzed after a full or partial frame has been recorded using a pre-determined real-time analysis pipeline specifically optimized to detect events of interest in the current sample. When an event is detected, confocal imaging is paused, and MINFLUX is applied. We used ROIs around and below $1 \times 1 \, \mu m^2$, and further implemented the framework to support dynamic determination of the ROI size and shape for more complex events. After the MINFLUX acquisition has finished, either after a pre-determined or flexible duration, the comprehensive dataset also allows the user to validate detected events during post-processing. etMINFLUX is implemented in a way to allow for indefinite acquisitions without manual interference, automatically returning to confocal monitoring after a finished event.

The etMINFLUX framework was implemented in an interactable standalone Python-based, and open-source control widget (Supplementary Fig. 1 and Supplementary Note 1), allowing experiment

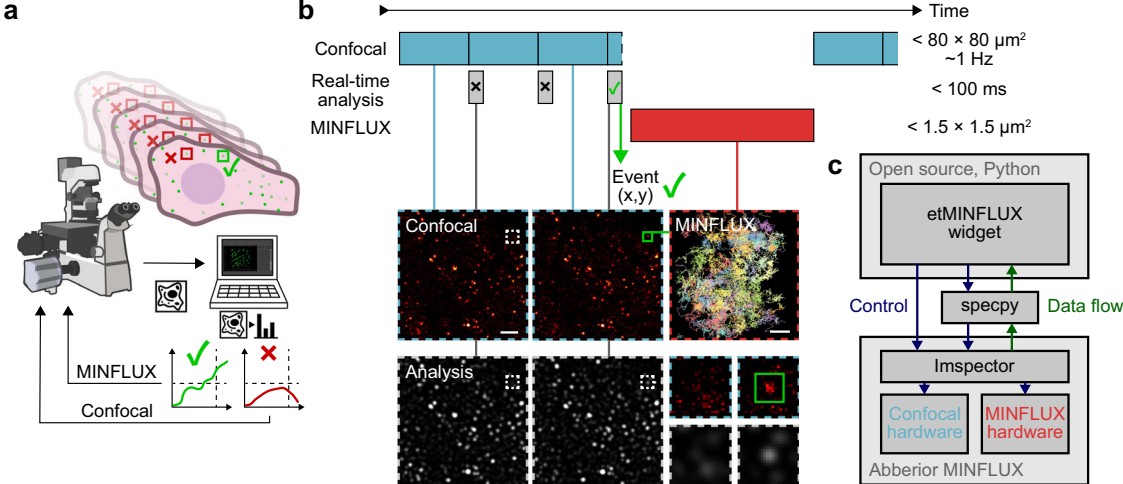

**Fig. 1 | Overview of event-triggered MINFLUX and software widget. a** Simplified sketch of the etMINFLUX method. **b** Timeline sketch and timings and sizes of an etMINFLUX experiment (top), with confocal imaging (top, blue), real-time analysis (middle, gray), and MINFLUX tracking or imaging (bottom, red). Example confocal images, processed analysis images, and a MINFLUX tracking dataset (bottom).

**c** Block diagram of microscope, microscope control software, and etMINFLUX control widget. Scale bars: 2 μm (confocal in **b**), 200 nm (MINFLUX in **b**). Microscope sketch in (**a**): NIAID Visual & Medical Arts, (10/7/2024), Super Resolution Fluorescence Microscopy, NIAID NIH BIOART Source, https://bioart.niaid.nih.gov/bioart/503.

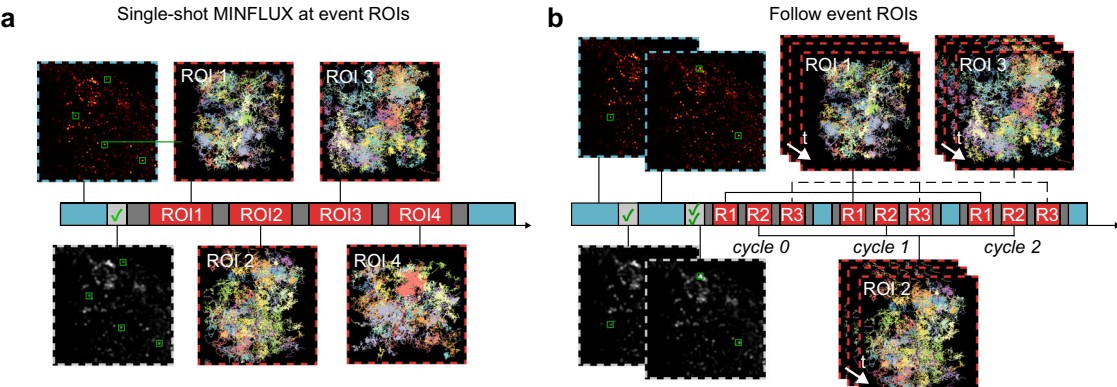

**Fig. 2 | Experimental modalities enabled by event-triggered MINFLUX.**
**a** Multiple event ROI recordings after detecting multiple event sites simultaneously.
**b** Multiple event ROI (R1, R2, R3) following after detecting multiple events in an event detection time window, with interleaved confocal and MINFLUX recordings of all sites. Timescales show confocal acquisition (blue), MINFLUX acquisition (red), real-time analysis (light gray) and overhead (dark gray). Confocal images are shown with a red-orange look-up table (top), and processed images from the analysis pipeline are shown with a gray look-up table (bottom). MINFLUX tracks are colored according to their track id. Scale bars: 2 μm (confocal in **a**, **b**), 200 nm (MINFLUX in **a**, **b**).

control and displaying rich event information to the user during an experiment. The widget interacts with the Imspector control software of the microscope through the specpy Python API (Fig. 1c). The framework builds on earlier work for event-triggered STED imaging[22], with a number of key differences requesting the employment of a newly developed widget: further development of the event-triggered concept, required control of high-resolution acquisition parameters, as well as practical factors such as the commercial software interaction and point-scanning monitoring in comparison with the earlier direct integration in a custom-written software and widefield-based monitoring (discussed in detail in Supplementary Note 2). The implementation is flexible to the type of image analysis to apply, implementing it as a general Python function and thus allowing the user to flexibly add new detectable event types. The framework also implements a simulation mode that can run without access to specpy or a microscope where previously recorded confocal timelapse data can be loaded and analysis pipeline parameters can be optimized (Supplementary Note 1).

Certain cellular processes that the method can investigate will benefit from extracting confocal and MINFLUX data in a timelapse and interleaved manner, allowing to follow the temporal development of both a dynamic confocal signal and the single-molecule tracks of the molecules of interest. In order to flexibly allow this, we developed four different recording modes (Fig. 2 and Supplementary Fig. 2). The simplest form of etMINFLUX is a single MINFLUX acquisition following the detection of a single event (Supplementary Fig. 2). In a second mode, multiple event sites are simultaneously identified and MINFLUX acquisitions are subsequently applied at ROIs surrounding each event site for a fixed amount of time (Fig. 2a). The third recording mode deals with following an event site for a prolonged time, where cycles of interleaved MINFLUX and confocal acquisitions are performed at a pre-set confocal frame rate following an event detection (Supplementary Fig. 2). In the fourth recording mode, multiple event sites can be detected over a time window, and are subsequently followed simultaneously in a cyclic manner (Fig. 2b and Supplementary Note 1). Such a recording mode is beneficial to increase the data throughput when the ROI sites are rare and should be followed for a longer time. The framework implementation of recording modes is readily extensible.

During any etMINFLUX experiment, the user is free to interact with the acquisition and can apply changes such as moving directly to the next ROI, moving to the next recording cycle, or deleting queued ROIs (Supplementary Fig. 1). To overlap confocal and MINFLUX data, for trustworthy further analysis, a residual relative shift between the coordinate systems of the two methods occasionally has to be applied depending on the confocal scanning speed (Supplementary Note 3, Supplementary Fig. 3). To facilitate the use of our framework implementation of the method, we developed a napari[33] widget that enables loading and visualization of combined confocal and MINFLUX data. The widget supports all etMINFLUX recording modes, including single-ROI and ROI-following acquisitions, as well as both 2D and 3D localization data (Supplementary Note 4).

Using this practical framework for event-triggered MINFLUX microscopy, we then tested the method across several cellular processes. We first applied it to quasi-static structures using simple recording modes and progressively increased the experimental complexity to investigate dynamic processes on timescales from tens of seconds to minutes. These experiments demonstrate both the power of etMINFLUX and the versatility of the implemented framework.

### etMINFLUX to measure lipid diffusion at caveolae

As a first step to showcase etMINFLUX in a relatively simple system, and to clarify key aspects of the technical implementation while obtaining a baseline measure of the temporal throughput enabled by the method, we applied the etMINFLUX framework to measure lipid diffusion at caveolae sites in PtK2 cells with high throughput (Fig. 3a). Caveolae are abundant plasma membrane invaginations of 60–80 nm in diameter of varying depth, mainly formed by Caveolins and Cavins[34,35]. They are known to be involved in a range of processes involving signal transduction, endocytosis, mechanoprotection, as well as lipid regulation[36], but many questions around their role still remain unanswered. Investigating nanoscale lipid composition and lipid diffusion at these sites can therefore provide mechanistic insight into their proposed role in lipid regulation. Perturbations of caveolae have also been linked to metabolic phenotypes and diseases, including altered insulin signaling[37].

While caveolae are dynamic structures, both in formation, disassembly, scission, and membrane movement, many are static over a timescale of minutes (Supplementary Fig. 4)[38]. Therefore, it served as an initial test for the method, both for the single ROI and multiple consecutive ROIs single-shot recording modes (Fig. 3b). Specifically, we expressed fluorescent-protein-tagged Caveolin1 (Caveolin1-EGFP) in live cells. Following a confocal image at a size of 15–80 × 15–80 μm², which took 0.3–3.2 s, the peak_detection_bright analysis pipeline (Supplementary Note 5, Supplementary Fig. 5a and Supplementary Table 1) was run to detect peaks of the Caveolin1-EGFP fluorescence signal, taking on average 69 ms (Supplementary Fig. 6a). With optimized pipeline parameters, the pipeline runs with a Caveolin1

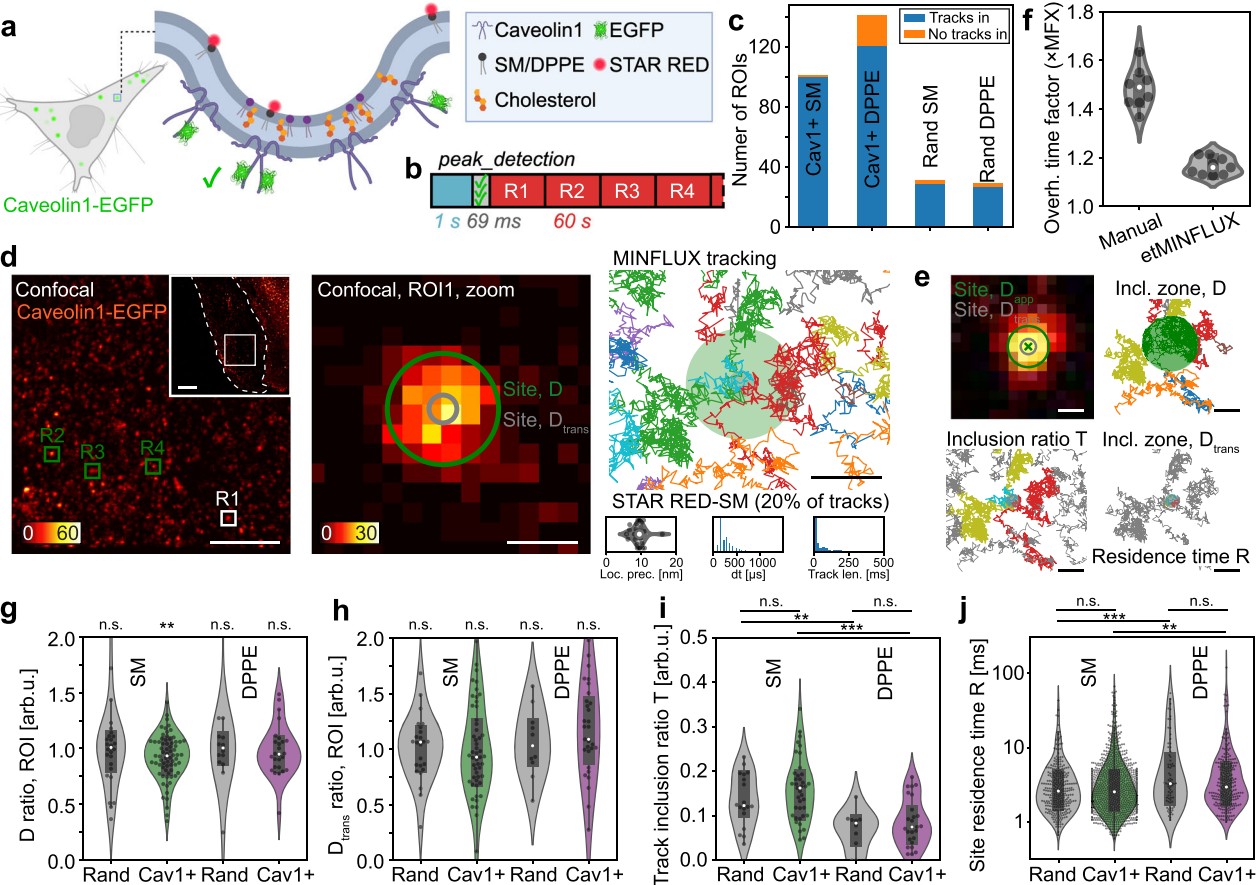

**Fig. 3 | Lipid dynamics measurements at caveolae sites following event detection of Caveolin1 sites. a** Sketch of caveola with Caveolin1-GFP (Cav1 + ) and STAR RED labeling of SM or DPPE. **b** etMINFLUX experiment timeline, with confocal (blue), analysis (gray), and MINFLUX (red) blocks, and median execution times over all experiments. **c** Number of ROIs for different conditions showing tracks inside the site (blue) or not (orange). Rand: random sites. **d** Exemplary experimental data, showing confocal image of Caveolin1-EGFP (left), zoom to event ROI (middle), and MINFLUX data overlaid on fitted caveola (right). MINFLUX metadata (localization precision, time between localizations, track length) populations for the shown dataset (bottom). Representative example of $n = 90$ events. **e** Visualization of analysis concepts: inclusion zones; localizations inside (green) diffusion inclusion zone; tracks passing (colored) inclusion ratio zone; and localizations inside (colored) transient diffusion and residence time inclusion zone. **f** Temporal overhead factor for manual and etMINFLUX event triggering. etMINFLUX: $n = 42$ ROIs, $n = 9$ cells, $n = 3$ experiments. Manual: $n = 80$ ROIs, $n = 9$ cells, $n = 3$ experiments. **g–j**. Analysis of diffusion coefficient ratio (**g**), transient diffusion coefficient ratio

(**h**), track inclusion ratio (**i**), and site residence time (**j**). Datapoints represent the mean of one caveola (**g**, **h**, **i**) or track (**j**). Random SM (gray): $n = 306$ tracks, $n = 26$ ROIs, $n = 3$ experiments; Cav1 + SM (green): $n = 1157$ tracks, $n = 69$ ROIs, $n = 5$ experiments; Random DPPE (gray): $n = 93$ tracks, $n = 17$ ROIs, $n = 3$ experiments; Cav1 + DPPE (magenta): $n = 264$ tracks, $n = 26$ ROIs, $n = 5$ experiments. **g** Rand SM: $p = 0.918$, Cav1 + SM: $p = 0.002$, Rand DPPE: $p = 0.624$, Cav1 + DPPE: $p = 0.688$. **h** Rand SM: $p = 0.575$, Cav1 + SM: $p = 0.217$, Rand DPPE: $p = 0.710$, Cav1 + DPPE: $p = 0.620$. **i** Cav1 + : $p = 0.00001$, Rand: $p = 0.002$, SM: $p = 0.110$, DPPE: $p = 0.187$. **j** Cav1 + : $p = 0.002$, Rand: $p = 0.000002$, SM: $p = 0.355$, DPPE: $p = 0.110$. Statistical tests: 1-sample Student's $t$ test for expected value = 1 (**g**, **h**); independent 2-sample Student's $t$ test for identical expected value (**i**, **j**). Scale bars: 10 μm (overview **d**), 5 μm (confocal **d**), 250 nm (ROI **d**; MINFLUX **d**; **e**). Subsampling: in **d**, MINFLUX tracks shows 20% of all acquired tracks. Violin plots: white point shows median, box spans IQR, whiskers extend 1.5 × IQR. Source data are provided in Source Data. **a** was created in BioRender, De Angelis, G. (2026) https://BioRender.com/4rus5v5.

accumulation detection precision of 1.0, recall of 0.73, and $F\beta_{0.25}$ score of 0.98 (Supplementary Note 6). As caveolae sites were detected, a fast 2D MINFLUX tracking acquisition with a triangle targeted coordinate pattern (TCP) of a dye-lipid conjugate of STAR RED and either sphingomyelin (SM) or 1,2-dipalmitoyl-sn-glycero-3-phosphoethanolamine (DPPE) was performed for around 60 s in a region of 1–2 × 1–2 μm², before either moving to the next detected caveola or back to the confocal for a new detection round. Resulting MINFLUX tracking metadata distributions for the temporal track length (62 ± 111 ms), time between localizations (345 ± 888 μs, minimum 86 μs, median 153 μs), and localization brightness (213 ± 102 kHz) for a representative complete raw dataset are plotted in Supplementary Fig 7a. In total, thanks to the automation, hundreds of ROIs surrounding individual caveolae and random control sites were measured (Fig. 3c). A vast majority (92%) of ROIs showed at least one lipid passing through the site as

analyzed with a diameter of 100 nm, indicating the power of limiting the acquisition to a small region to rapidly sample the measured space: acquiring a larger area spanning multiple ROIs would lead to a significantly lower throughput of the data of interest and would increase the risk of the site dynamically moving during MINFLUX acquisition due to longer recording times.

Single or multiple ROIs were acquired in the same cell sequentially after a single confocal detection (Fig. 2a, Supplementary Fig. 2 and Supplementary Movie 1), and the confocal and MINFLUX data was overlaid (Fig. 3d). The recording of a secondary confocal image post-MINFLUX and fitting of the precise caveolae site position allows exclusion of moving event sites. The rich data from the etMINFLUX recordings allows flagging lipid tracks and track segments that were present at the site, and further lipid diffusion analysis with high statistical precision thanks to the high throughput (Fig. 3e and

Supplementary Figs. 8–10). The exemplary MINFLUX dataset was recorded with a mean localization precision of 9.2 nm, median time between localizations of 220 μs, and mean track length of 56 ms (Fig. 3e, bottom). To quantify the improvement in temporal throughput provided by etMINFLUX, we performed manual acquisition experiments mimicking the etMINFLUX workflow and calculated temporal overhead factors (Supplementary Note 7). Manual acquisitions exhibited an average overhead factor of 1.49 ×, whereas etMINFLUX reduced this to 1.17 × (Fig. 3f), corresponding to ~10 s of additional time per ROI compared to ~30 s for manual operation (Supplementary Fig. 11a).

We performed lipid diffusion analysis in multiple steps. We first applied filtering steps to the MINFLUX tracking data to retain high-quality single-particle trajectories (Supplementary Note 8 and Supplementary Fig. 8). Then, square displacement analysis and fitting of the resulting datasets resulted in diffusion coefficients for whole and segments of lipid tracks (Supplementary Note 9) as well as dynamic 2D localization precisions, determined on average to be 9.5 nm (Supplementary Fig. 12). Using only the tracks that at some point passed through a marked site (as defined by a circle with a radius of 200 nm; Supplementary Note 9–10, Supplementary Fig. 13), we divided these tracks into continuous segments that stay either entirely inside or outside the caveola and extracted a diffusion coefficient for each segment. For each site, we extracted a mean diffusion coefficient for the segments inside and outside (Supplementary Fig. 14) and calculated the diffusion coefficient ratio as the diffusion coefficient inside divided by the diffusion coefficient outside. We compared the results from random non-caveolae sites, where we expect a ratio of 1, with those from caveolae, for SM and DPPE (Fig. 3g). The ratios at random sites were indeed centered at 1 (SM: $1.01 \pm 0.07$, $p = 0.918$; DPPE: $1.06 \pm 0.12$, $p = 0.624$). At caveolae, the ratios for SM showed a significant decrease of diffusion speed inside the caveolae ($0.92 \pm 0.02$, $p = 0.002$), while DPPE did not show a difference ($0.98 \pm 0.04$, $p = 0.688$). This indicates an inhibition of diffusion of SM in caveolae, while DPPE was not significantly affected compared to elsewhere on the membrane.

Visual inspection of the trajectories suggested the presence of confined diffusion segments, indicative of hop or trap diffusion. To investigate this further, we performed packing coefficient analysis[39] to segment confined portions of the tracks and analyze their occurrence and spatial distribution (Supplementary Fig. 15). The results showed that DPPE contained a higher fraction of tracks with confined segments (47%) than SM (39%) and, on average, spent a longer time in confined states. By extracting the spatial positions of confined segments, we derived radial profiles of the temporal confinement fraction, which is defined as the fraction of time spent in confined states at a given radial distance. This analysis revealed that DPPE exhibited more frequent confinement than SM at most radial positions, except within the caveolae region ($d_r < 50$ nm), where both lipids showed similar confinement fractions (~0.18). Restricting the analysis to tracks that passed through the event site, or those that did not, did not reveal any striking differences.

We further calculated the transient diffusion coefficient for each localization (sliding window of 100 localizations; Supplementary Note 9). For each caveola we extracted the average diffusion coefficient for localizations inside and outside the site (Supplementary Fig. 14), as segmented with a diameter of 100 nm, and again extracted a ratio between the two (Fig. 3h). The expected ratio around 1 was observed for random sites with SM ($1.01 \pm 0.06$, $p = 0.575$) and DPPE ($1.06 \pm 0.09$, $p = 0.710$), while there was a non-significant trend at caveolae towards lower ratios for SM ($0.95 \pm 0.06$, $p = 0.217$) but not DPPE ($1.05 \pm 0.14$, $p = 0.620$).

To investigate the involvement of the lipids at sites, we analyzed the track inclusion ratio $T$ by taking the ratio between the number of tracks in an ROI that pass through the caveola and the number of

tracks that do not pass through (Fig. 3i). The larger the ratio, the more the lipid was, in theory, enriched in the measured caveola. For the selected caveolae we observed a higher inclusion ratio $T$ of SM compared to DPPE (SM: $0.16 \pm 0.01$, DPPE: $0.09 \pm 0.01$; $p = 0.00001$), i.e., an enrichment of SM in caveolae compared to DPPE. The control measurements on random (non-caveolae) sites revealed a decreasing trend of the inclusion ratio compared to the caveolae sites for both lipid analogs (SM, random: $0.14 \pm 0.01$, $p = 0.110$; DPPE, random: $0.07 \pm 0.01$, $p = 0.187$) and a higher inclusion ratio of SM compared to DPPE ($p = 0.002$). This results from the fact that diffusion of SM and DPPE is in general, even outside caveolae, different (Supplementary Fig. 9–10, Supplementary Movie 1–2)[40–42]. Lastly, we investigated the site residence times R for track segments that passed through the caveola, extracting the time between the first localization upon entry and the last localization upon leaving the site for a 100 nm diameter (Fig. 3j). Here, we observed a mean $R$ of around 4–8 ms (SM random: $3.84 \pm 0.26$ ms; SM caveolae: $4.17 \pm 0.17$ ms; DPPE random: $7.79 \pm 1.26$ ms; DPPE caveolae $5.61 \pm 0.63$ ms). DPPE did stay longer in caveolae ($p = 0.002$) and random sites ($p = 0.000002$) than SM, which cannot be explained by slower diffusion rates (Supplementary Fig. 14). Most importantly, neither SM ($p = 0.355$) nor DPPE ($p = 0.110$) resided significantly longer in caveolae than at random sites.

## etMINFLUX 3D tracking topography of budding endosomes in live cells

As a second biological system, we applied etMINFLUX to a spatially random, rare, and rapid cellular process occurring on the seconds scale. Specifically, we used 3D MINFLUX tracking of a cholesterol-derived membrane probe (STAR RED-membrane) to measure membrane topography at Dynamin1-accumulation sites in HeLa cells (Fig. 4a). Dynamin1 is known to accumulate and act as the initiator of constriction and scission at clathrin-coated pits, caveolae, and other endocytosis sites[43], where cholesterol further is important for membrane curvature[44] and thus abundantly present. The morphology of pre-scission endocytic vesicles, such as neck length and vesicle diameter, as well as their temporal progression and arrest are key signatures that define vesicle type. These parameters can reflect upstream states of membrane tension, lipid composition, and actin-dependent forces[45,46], and are also altered in disease models, including those involving Dynamin1 mutations[47]. This highlights the need to measure vesicle morphology and progression dynamics in living systems with the highest possible precision. The process of accumulation of Dynamin, constriction, and scission is a fast process with respect to MINFLUX acquisitions, occurring on a time scale of tens of seconds[48,49]. Here, we highlight that etMINFLUX was able to catch a process in living cells not feasible to catch with high throughput with normal MINFLUX microscope control and acquisitions due to its rarity and rapidity – in this case, pre-scission budding endosomes. Capturing such events and structures in living cells, in 3D, with high throughput, and with the localization precision enabled by MINFLUX, opens new possibilities for analyzing their structure and dynamics. Previously, such studies were largely limited to electron microscopy[50] or to 2D when using fluorescence microscopy in living cells[51]. We performed 3D tracking with an octahedral TCP of STAR RED-membrane, hypothesized to be incorporated in the site due to the local cholesterol-enriched membrane, in order to get a 3D topographic map. Resulting MINFLUX tracking metadata distributions for the temporal track length ($46 \pm 89$ ms), time between localizations ($485 \pm 626$ μs, min 357 μs, median 359 μs), and localization brightness ($172 \pm 80$ kHz) for a representative complete raw dataset are plotted in Supplementary Fig. 7b. We used the single ROI follow mode, to ensure the fastest sampling with MINFLUX of a single site (Fig. 4b and Supplementary Movie 3–5). Following confocal images of a ~15 × 15 μm² area, with frame rates of around 1 Hz, the dyn_signalrise real-time analysis pipeline (Supplementary Note 5, Supplementary Fig. 5b and Supplementary Table 1) was applied, taking

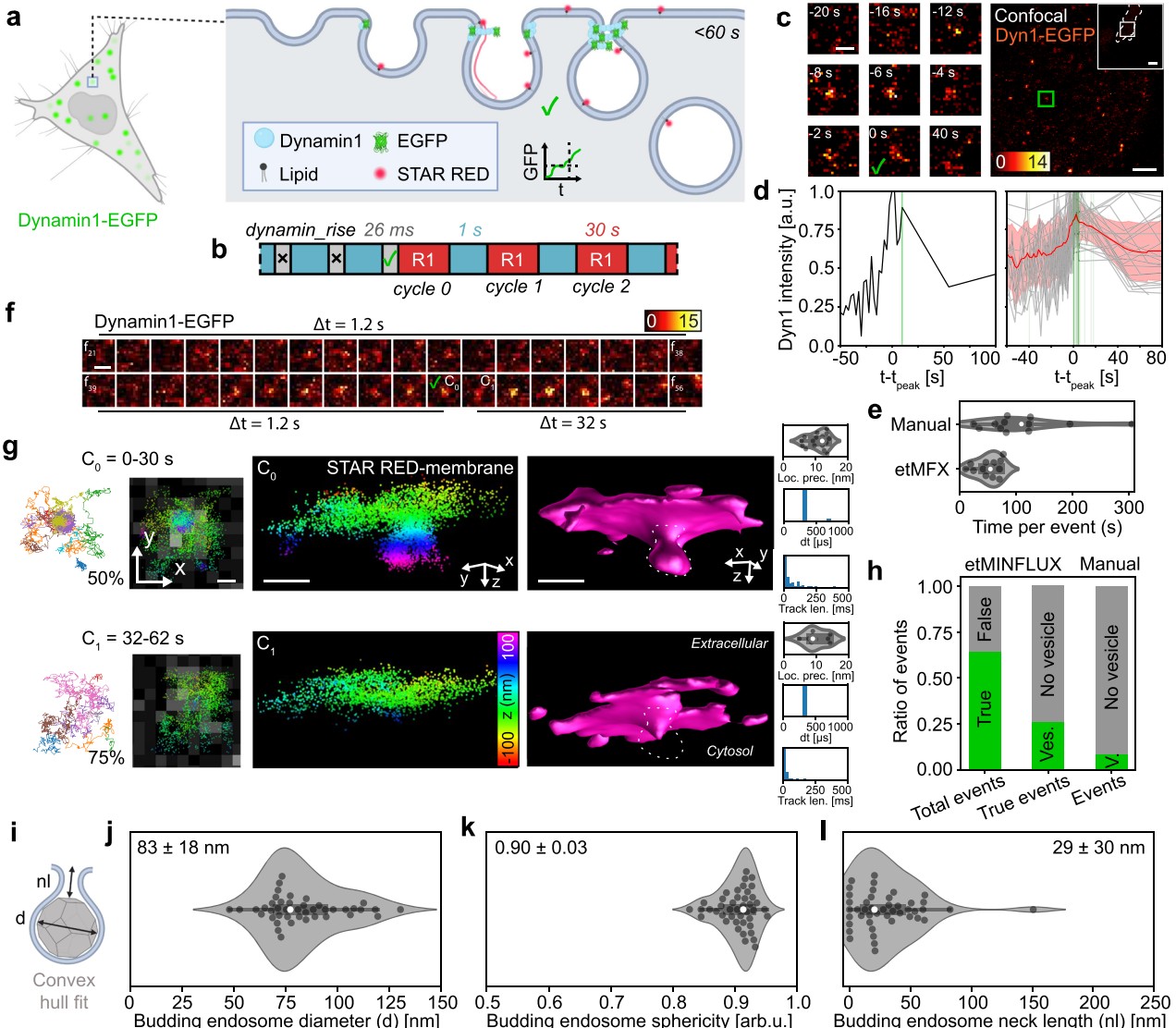

**Fig. 4 | Live imaging of budding endocytic vesicles in 3D following event detection of Dynamin1 accumulation. a** Sketch of Dynamin1-EGFP accumulation and budding and lipid labeling with STAR RED-membrane. **b** etMINFLUX experiment timeline, with confocal (blue), analysis (gray), and MINFLUX (red, R#) blocks, and median execution times over all experiments. **c** Event detection example with Dynamin1-EGFP accumulation in zoom-in. Check mark marks event triggering. **d** Exemplary (left) and all (right) Dynamin1 intensity event detection curves, with mean ± SD curve (red). Green lines indicate event detection. **e** Monitoring time per detected event for manual and etMINFLUX event triggering. etMINFLUX: $n = 79$ events, $n = 13$ cells, $n = 4$ experiments. Manual: $n = 75$ events, $n = 13$ cells, $n = 3$ experiments. **f** Exemplary confocal time lapse (frames 21–56) zoom of event area at 1/1.2 Hz or 1/32 Hz before or after event detection. $C_i$ indicates MINFLUX recording cycle i. **g** Exemplary MINFLUX data of event in (**f**) showing STAR RED-membrane MINFLUX tracks and localizations overlaid on confocal zoom (left), z-color-coded 3D localization point cloud (middle), and surface fitting of MINFLUX data (right) with endosome highlighted (dashed line). Data is shown for cycle 0 and cycle 1 of the same event. MINFLUX metadata (localization precision, time between localizations, track length) populations for the shown datasets (right). Representative example of $n = 43$ events with a budding endosome. **h** Post-experimental statistics of true event detections and number of vesicles visible in etMINFLUX and manual MINFLUX data. **i** Endosome convex-hull-fitting sketch with diameter (d) and neck length (nl). **j–l** Budding endosome diameter (**j**), sphericity (**k**), and neck length (**l**) analysis. Datapoints represent endosomes before scission in single cycles of etMINFLUX data in true event detections. $n = 43$ events, $n = 23$ cells, $n = 4$ experiments. Scale bars: 10 μm (overview **c**), 2 μm (confocal **c**), 500 nm (ROI zoom **c**; **f**), 100 nm (**g**). Subsampling: in (**g**) 2D MINFLUX tracks shows 50% or 75% of all tracks from MINFLUX acquisition. Violin plots: white point shows median, box spans IQR, whiskers extend 1.5 × IQR. Source data are provided in Source Data. **a** was created in BioRender, De Angelis, G. (2026) https://BioRender.com/ah6nktp. **i** was created in BioRender, De Angelis, G. (2026) https://BioRender.com/rzieic9.

on average 26 ms (Supplementary Fig 6b). Confocal imaging was continued until a Dynamin1 accumulation event was detected, at which point the system switched to MINFLUX 3D tracking of the fluorescent cholesterol-based membrane probe with recording windows of ~20 s in ~0.5 × 0.5 μm² ROIs.

A typical event detection is shown in Fig. 4c. Here, a 15 × 15 μm² ROI was continuously recorded, and an event was detected at timepoint $t = 0$, where the Dynamin1 signal was continuously increasing

(Fig. 4d). Following event detection, after the first MINFLUX acquisition cycle, the Dynamin1 signal had dropped, indicating Dynamin1 having dispersed again and an endosome likely having been released. The mean of all temporal Dynamin1 signal curves at detected events (bold red line in Fig. 4d) indicated a rapid signal increase in the 20 s leading up to the peak Dynamin1 signal, after which the signal more slowly decayed as the Dynamin1 constriction was disassembled and the protein dispersed again. The full process finished in < 60 s. Event

detections were mostly done at least 5 s before the peak detected signal (green vertical lines in Fig. 4d), and likely more before the actual Dynamin1 accumulation peak due to the decreased confocal sampling of the Dynamin1 signal after event detection. This indicated the possibility to most commonly catch an endosome before scission in the first MINFLUX acquisition cycle.

To quantify event detection efficiency and data throughput in this application, we performed manually triggered experiments mimicking the etMINFLUX workflow. While monitoring a continuously running confocal acquisition, events were identified manually, and MINFLUX acquisitions were initiated also manually at the detected sites (Supplementary Note 7). We measured the total experiment time and calculated the average time between the end of one MINFLUX acquisition and the detection of the next event. etMINFLUX reduced this interval by approximately half, from 109 s to 50 s (Fig. 4e), highlighting the difficulty of reliably detecting such rapid events through manual control. In some cases, etMINFLUX required as little as ~10 s between successive events. We further analyzed the throughput of useful data, defined as acquisitions capturing a Dynamin1 event together with an overlapping endocytic vesicle in the MINFLUX data. Here, etMINFLUX improved the success rate by a factor of three, capturing vesicles in 26% of events as compared to only 9% with manual control (Fig. 4h and Supplementary Note 7). Overall, this corresponds to an approximately sixfold increase in useful data throughput when using etMINFLUX to study a rapid, seconds-scale cellular process such as endosomal budding.

Visualizing the data for an exemplary event caught with etMIN-FLUX, we noted a continuously increasing Dynamin1-EGFP signal appearing from a background in the confocal data, and after some time following event detection, the signal decreased and disappeared, indicating a scission event (Fig. 4f and Supplementary Fig. 16). The recorded MINFLUX data corroborated the confocal data, showing a budding endosome at a late stage in the first recording cycle of 30 s (Fig. 4g, top) but a flatter membrane without an endosome head already in the next recording cycle (Fig. 4g, bottom; Supplementary Movie 3). The MINFLUX datasets in the two cycles were recorded with a mean localization precision of 10.6 nm and 10.1 nm, median time between localizations of 359 μs and 359 μs, and mean track length of 58 ms and 31 ms, respectively (Fig. 4g, right). The presence of Dynamin1-EGFP signal for another three recording cycles may indicate that the budding endosome was either still attached but so restricted that no STAR RED-membrane probe could diffuse into it during the following recording cycles, or that the Dynamin1 took an unusual long time to diffuse away from the site. We measured many similar events, where a majority showed temporally overlapping Dynamin1-EGFP signal disappearance and 3D MINFLUX topography budding endosome disappearance (Supplementary Fig. 16, 17 and Supplementary Movie 4).

In total, the live etMINFLUX event detection reached a precision of 65% according to post-experimental analysis (Fig. 4h). False positives are often rapidly moving peaks that are difficult to track without temporal delays due to the long frame-to-frame distances, thus appearing as novel peaks in the backend tracking. When run in an offline automated pipeline optimization routine with pipeline parameters optimized towards precision, the analysis pipeline is nevertheless capable of reaching a mean precision of 1.0, while having a recall of 0.18 and an $F\beta_{0.25}$ score of 0.72 (Supplementary Note 6 and Supplementary Fig. 18a). However, while the etMINFLUX experiments ran with a lower precision, post-experimental analysis allowed straightforward scrutinization of events thanks to the rich information and data saved for each event. Of the real events detected, only a minority of 26% showed the presence of budding endosomes in the MINFLUX data overlapping with the confocal Dynamin1 signal. This may be explained in different ways: (1) due to the sparse labeling necessary for MINFLUX, the fluorescent membrane probe did

occasionally not enter a budding endosome present either by chance or by exclusion before scission; (2) an event was detected at a late stage of the budding and the MINFLUX acquisition was thus started too late, i.e., after endosome scission occurred; or (3) no budding vesicle was ever present. Nevertheless, the smart method allows high-throughput 3D measurements of naturally occurring endosomal events, which otherwise is difficult considering its rarity and difficulty to measure.

We further confirmed the identity of these vesicles as endocytic vesicles and demonstrated the specificity of etMINFLUX by performing control experiments in which Dynamin1-negative event sites were selected. These were randomized from binary maps excluding the Dynamin1 signal using functionality implemented in the framework, and compared with Dynamin1-positive sites (Supplementary Fig. 19). Each control site was followed for 1.5–6.5 min over multiple MINFLUX acquisition cycles. In these experiments, only 9.5% of the 42 confirmed Dynamin1-negative sites showed any endocytic-like vesicle within the followed ROI, compared to 39% for confirmed Dynamin1-positive events (Supplementary Fig. 19a). Importantly, none of the control events showed vesicles overlapping with Dynamin1 signal.

A majority (54%) of budding endosomes visible in the MINFLUX data were present only in the first acquisition cycle (20–30 s), occasionally (15%) the same budding endosomes were present also in the second cycle (~40–60 s) or in the third or further cycles (15%) (~60 + s) before seemingly undergoing scission. Rarely (15%) an identified endosome stayed for many acquisition cycles and therefore minutes following an event detection, which then indicated an arrested endosomal budding state with Dynamin1 accumulation (Supplementary Fig. 17, Supplementary Movie 5).

The MINFLUX topographic mapping of the captured Dynamin1-associated budding events allowed us to accurately measure the nanoscale 3D geometry of pre-scission endocytic vesicles. For this, the localizations representing the endosomal surface were recognized and separated from the rest of the localization cloud using dedicated filtering steps, a convex hull was fitted to the remaining localizations, and the average position of the membrane was determined through a Gaussian fitting of the z-profile (Fig. 4i and Supplementary Note 11, 12). From this, the endosomal equivalent spherical diameter (Fig. 4j), sphericity (Fig. 4k), and neck length (Fig. 4l) was extracted (Supplementary Note 11). The endosomal equivalent spherical diameter spanned 50–130 nm, with a mean size of 83 ± 18 nm, matching values derived from electron microscopy[48]. The sphericity showed almost exclusively nearly spherical endosomal heads, with a mean sphericity value of 0.90 ± 0.03. Lastly, the neck length was measured to be 29 ± 30 nm on average, with a large span of values also comprising an example of an endosome with a neck length of 151 nm (Supplementary Fig. 20).

## etMINFLUX 3D tracking measures membrane fluidity at Gag accumulation sites

To further prove the applicability of the system on minutes-scale cellular processes, we next tested our etMINFLUX 3D tracking framework to monitor molecular membrane diffusion and organization at viral budding sites, taking Gag proteins as an example. Gag proteins are essential structural proteins for viruses such as retroviruses and specifically HIV. They assemble into membrane-bound aggregates from individual subunits in the early stages of virus budding at the plasma membrane of infected host cells, which is a driving factor in the bending of the membrane[52]. During the process, a uniquely composed viral membrane is formed[53,54], crucially determining the infectivity of the virus[55,56]. Enabling temporal high-resolution 3D measurements from the onset of budding of the membrane topography and lipid dynamics can lead to insights on the underlying viral-forming mechanisms and how this can be used for drug development. We used etMINFLUX with 3D MINFLUX tracking with an octahedral TCP of the fluorescent cholesterol-based membrane probe STAR RED-

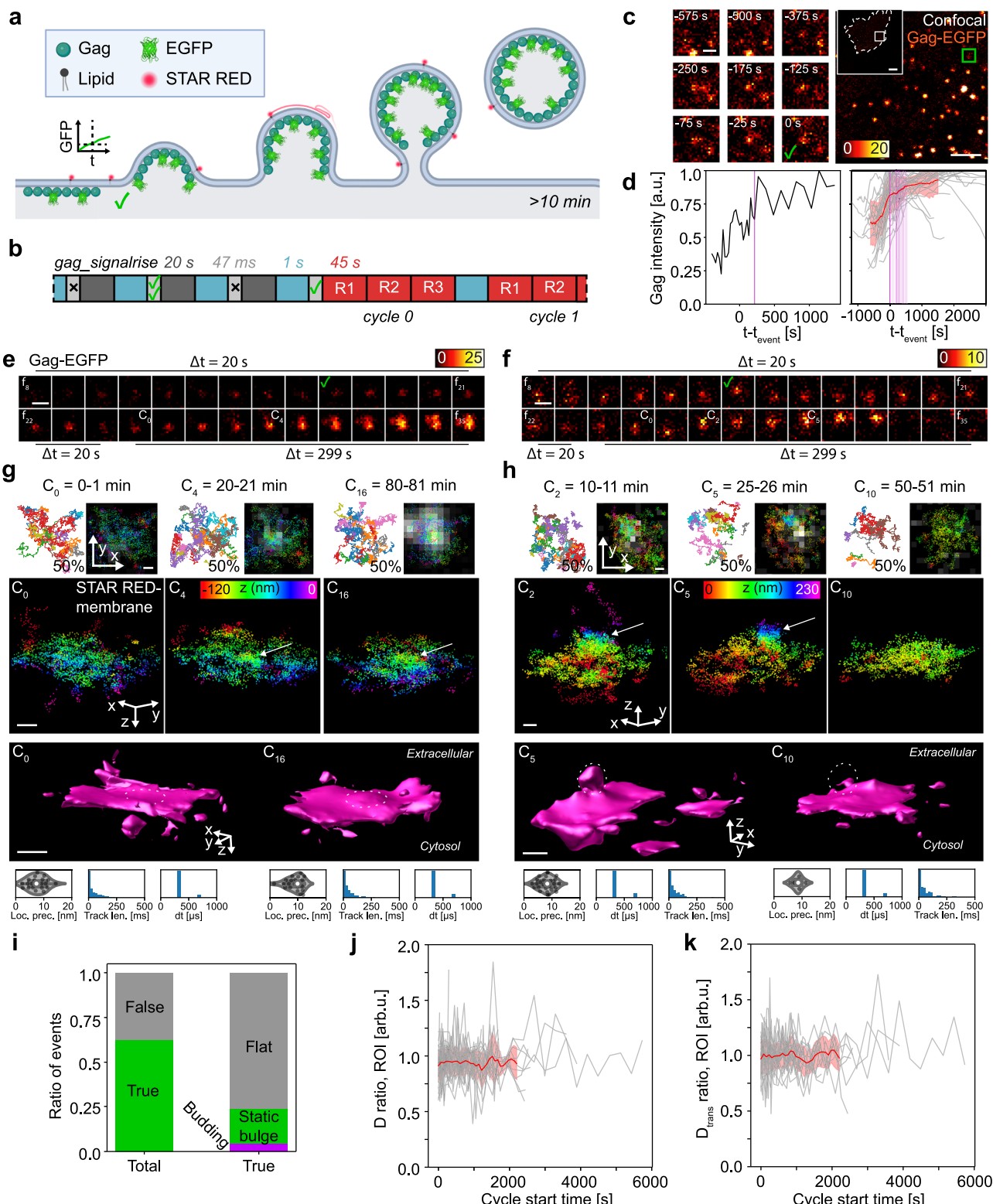

membrane to investigate the membrane shape and fluidity over time during the Gag accumulation process in live HeLa cells (Fig. 5a). Gag accumulation leading to virus formation and budding is a slower process than endosome budding, where the accumulation is known to take place over approximately 10 min[57,58], and the full process to the time of release around 25 min[59]. As such, we applied either the single ROI follow or the multiple ROI follow mode to follow the evolution of Gag accumulation sites in both the confocal and MINFLUX channels in multiple recording cycles (Fig. 5b and Supplementary Movie 6). We

used a lower confocal frame rate around 1/20 Hz in an area of $10-25 \times 10-25\ \mu m^2$. The gag_signalrise real-time analysis pipeline (Supplementary Note 5, Supplementary Fig. 5c and Supplementary Table 1), taking on average 47 ms (Supplementary Fig. 6c), was applied to detect the accumulation of Gag. MINFLUX data was acquired with recording windows of 30–60 s in ~ $0.7 \times 0.7\ \mu m^2$ ROIs. The measurements were done either at room temperature or at 33–37 °C. We present the results altogether, and the diffusion analysis is always presented as ratio values between diffusion coefficients instead of

**Fig. 5 | Live imaging of Gag accumulation sites and membrane fluidity in 3D following event detection of Gag accumulation. a** Sketch of Gag-EGFP accumulation and budding and lipid labeling with STAR RED-membrane. **b** etMINFLUX experiment timeline, with confocal (blue), analysis (light gray), wait (dark gray), and MINFLUX ROI (red, R#) blocks, and median execution times over all experiments. **c** Event detection example with Gag-EGFP accumulation in zoom-in. Check mark marks event triggering. **d** Exemplary (left) and all recorded (right) Gag intensity event detection curves, with mean ± SD curve (red). Magenta lines indicate start times of the first MINFLUX cycle. **e, f** Exemplary confocal time lapse (frames 8–35) zooms of event areas at 1/20 Hz or 1/299 Hz before or after event detection. $C_i$ indicates MINFLUX recording cycle i. **g, h** Exemplary MINFLUX data of events shown in (**e, f**) from cycles 0, 4, 16 (left) or cycles 2, 5, 10 (right), showing STAR RED-membrane MINFLUX tracks and localizations overlaid on confocal zooms (top), z-color-coded 3D localization point clouds (middle), and surface fittings of MINFLUX data (bottom) with areas of bulging (**g**) and budding (**h**) highlighted (dashed line). MINFLUX metadata (localization precision, time between localizations, track length) populations for the shown datasets (bottom). **i** Post-experimental statistics of true event detections and number of dynamically budding (magenta) or bulging (green) membranes visible in 3D MINFLUX data. **j, k** Analysis of diffusion coefficient ratio (**j**) and transient diffusion coefficient ratio (**k**) over time since MINFLUX initiation for all followed events (gray) and their mean ± SD curve (red). n = 54 ROIs from n = 10 experiments. Scale bars: 10 μm (overview c), 2 μm (confocal c), 500 nm (ROI zoom c, e, f), 100 nm (**g, h**). Subsampling: in (**g, h**), 2D MINFLUX tracks shows 50% of all tracks from MINFLUX acquisition. Violin plots: white point shows median, box spans IQR, whiskers extend 1.5 × IQR. Source data are provided in Source Data. **a** was created in BioRender, De Angelis, G. (2026) https://BioRender.com/f6vehwf.

**Table 1 | MINFLUX sequence parameter summary for the three sequences used in this work: 2D tracking for Caveolin1 experiments, 3D tracking for Dynamin1 experiments, and 3D tracking for Gag experiments**

| Iteration | Pattern | TCP $L$ (nm) | Photon limit | Dwell time (μs) | Pattern repeats | CFR limit | Background threshold (kHz) | Laser power factor |
|---|---|---|---|---|---|---|---|---|
| | | | | seqTrk2D-cav-SR: Caveolae (Caveolin1), 2D tracking, SR-SM and SR-DPPE | | | | |
| 0 (Preloc.) | Hexagon | – | 20 | 500 | 1 | – | 15 | 1 |
| 1 | Hexagon | 302 | 10 | 100 | 1 | 0.5 | 30 | 1.5 |
| 2 | Hexagon | 151 | 6 | 100 | 1 | – | 30 | 2 |
| 3 | Triangle | 100 | 15 | 50 | 1 | – | 74 | 3 |
| | | | | seqTrk3D-dyn-SRmemb: Endocytosis (Dynamin1), 3D tracking, SR-membrane | | | | |
| 0 (Preloc.) | Hexagon | 288 | 40 | 500 | 1 | – | 30 | 1 |
| 1 | z-line | 288/1440 | 300 | 2000 | 1 | – | 30 | 1 |
| 2 | Octahedron | 288 | 40 | 200 | 1 | – | 40 | 1 |
| 3 | Octahedron | 151 | 30 | 200 | 1 | 0.9 | 55 | 2 |
| 4 | Octahedron | 100 | 20 | 200 | 1 | – | 71 | 3 |
| | | | | seqTrk3D-gag-SRmemb: Virus budding sites (Gag), 3D tracking, SR-membrane | | | | |
| 0 (Preloc.) | Hexagon | 288 | 40 | 500 | 1 | – | 30 | 1 |
| 1 | z-line | 288/1440 | 300 | 2000 | 1 | – | 30 | 1 |
| 2 | Octahedron | 288 | 40 | 200 | 1 | – | 40 | 1 |
| 3 | Octahedron | 151 | 30 | 200 | 1 | 0.9 | 50 | 2 |
| 4 | Octahedron | 100 | 20 | 200 | 1 | – | 60 | 3 |

absolute values to allow comparability in the broad range of the observed absolute values of diffusion coefficients (Supplementary Fig. 21). The average localization precision was determined to be 8.4 nm in these experiments (Supplementary Fig. 12). Notably, the mean 3D localization precision is better than that of 2D measurements. This arises from the sequence parameters used, specifically a higher photon limit while maintaining an identical $L$ size of the TCP (see Table 1 and Methods section for sequence parameters). The distributions of values of the temporal track length (51 ± 84 ms), time between localizations (545 ± 873 μs, min 357 μs, median 359 μs), and localization brightness (182 ± 90 kHz) obtained from the MINFLUX metadata of a representative complete raw dataset are plotted in Supplementary Fig. 7c.

Figure 5c depicts a typical event detection. A ROI of 10 × 10 μm² was here recorded at a rate of 1/25 Hz, and an event was detected at timepoint t = 0 after a slow and steady increase of the Gag signal over hundreds of seconds (Fig. 5d). Following the event detection and the start of MINFLUX acquisition, the confocal Gag signal kept increasing, indicating an early event detection well before the peak of Gag accumulation. This general trend in the Gag signal over time is exemplified by the mean of all temporal Gag signal curves (bold red line in Fig. 5d), where the Gag signal keeps increasing well into 30 min following event detections for most cases. In some cases, the Gag signal decreased within the 10–20 min following an event detection.

Two of the developing events are shown in Fig. 5e, g and Fig. 5f, h, respectively. One of these events, which was recorded at the basal membrane of the cell, showed a continuously increasing Gag signal (Fig. 5e), while the other event, recorded at the apical membrane, showed accumulation of Gag signal until a point of sudden disappearance (Fig. 5f). The MINFLUX data of the first event was recorded over a total of 20 recording cycles spaced 5 min apart, and in 3D initially showed a flat membrane which then continuously bulged with an increasing size (Fig. 5g, Supplementary Fig 16 and Supplementary Movie 7). The MINFLUX datasets in cycle 0 and cycle 16 were recorded with a mean localization precision of 7.7 nm and 8.8 nm, median time between localizations of 359 μs and 359 μs, and mean track length of 46 ms and 41 ms, respectively (Fig. 5g and Supplementary Fig. 16, bottom). The latter event with a suddenly disappearing Gag signal instead showed a budding and scission process, where cycles 0–2 highlighted the start to bulging (Supplementary Fig. 22a and Supplementary Fig. 16), cycle 5 showed a fully formed virus site with a narrowed neck, and cycles 7 and later, when the Gag signal had disappeared, showed a flat membrane (Fig. 5h, Supplementary Fig. 16 and Supplementary Movie 8). The MINFLUX datasets in cycle 2 and cycle 10 were recorded with a mean localization precision of 8.0 nm and 8.5 nm, median time between localizations of 359 μs and 359 μs, and mean track length of 64 ms and 50 ms, respectively (Fig. 5h, bottom). In total, the majority of Gag accumulation events were recorded at the bottom membrane and did not indicate any signs of bulging,

budding, or Gag signal disappearance linked to virus particle scission (Supplementary Fig. 22b). A minority of events showed signs of static bulging (Supplementary Fig. 22c and Supplementary Fig. 16) and only very few showed developing budding over time (Supplementary Fig. 22d and Supplementary Fig. 16).

Post-acquisition analysis of the data indicated that we reached an event detection precision of 63%, while the false positives mainly consisted of fluctuations in the Gag signal that disappeared quickly afterwards, such as from Gag accumulation site movement (Fig. 5i). Offline optimization of the parameters on pre-recorded confocal timelapses yielded similar optimal performance, with a mean precision of 0.45, a mean recall of 0.63, and an $F\beta_3$ score of 0.67 (Supplementary Note 6 and Supplementary Fig 18b), with recall prioritized during optimization. These values, which varied substantially between individual timelapses due to the small number of events present, highlight the difficulty of detecting the slow and dim accumulation of Gag signal and validate the performance achieved in the etMINFLUX experiments.

Of the true detected events, a clear majority of 76% showed a flat and non-changing membrane throughout the recorded cycles, despite an increasing Gag signal, while only 24% showed consistent bulging at the membrane. A great majority of these showed no change over recording cycles, and only 3 detected events rendered signs of budding, here defined as a membrane bulge dynamically increasing over time. We further confirmed that the observed membrane bulging and budding correspond to virus budding sites by performing control etMINFLUX experiments in which Gag-negative event sites were selected, and compared with Gag-positive sites (Supplementary Fig. 23), following the same procedure used for the Dynamin1-negative controls described above. The control sites were followed for 12–48 min over multiple MINFLUX acquisition cycles. As in the Gag-positive recordings, the membranes displayed varying degrees of flatness, bending, and folding. However, none of the 41 Gag-negative sites showed any consistent and localized outward membrane bending within the followed ROIs (Supplementary Fig. 23a), in contrast to the Gag-positive sites. To estimate the achievable data throughput for such events, we also performed manual acquisition experiments mimicking the etMINFLUX workflow. In these experiments, we detected only a single event site, which showed no consistent bulging or budding despite being followed for 35 min (Supplementary Fig. 24). This highlights the difficulty of detecting slow, minutes-scale intensity increases at an early stage using manual monitoring. Moreover, even when events are detected manually, aligning multiple events on a common temporal axis would be severely limited due to the difficulty of identifying the onset of the process and the lack of comprehensive time-resolved data recorded during etMINFLUX experiments.

We further investigated the general membrane fluidity at the Gag accumulation sites, using the diffusion of the cholesterol-derived STAR RED-membrane as a measure. We defined a diffusion coefficient ratio as the ratio between the diffusion coefficient inside and outside of the site as defined by a circle with radius 200 nm (Supplementary Note 9 and Supplementary Fig. 21), calculated its value over all MINFLUX recording cycles for all true detected event sites, and aligned the resulting values to the start of each first cycle MINFLUX recording (Fig. 5j). The value of the ratio was on average around 0.9 and hardly changed over time, indicating a slightly lower fluidity at the Gag accumulation sites than outside of them. We also defined a transient diffusion coefficient ratio, which was calculated over all MINFLUX cycles as the ratio between the transient diffusion coefficient inside the site, as defined by a circle with radius 60 nm, and outside of a larger surrounding area, as defined by a circle with radius 240 nm (Fig. 5k, Supplementary Fig. 8, Supplementary Note 9 and Supplementary Fig. 21). Compared to the diffusion coefficient ratio, the transient diffusion coefficient ratio thus compares transient diffusion between a more centered and more distant area. Neither this ratio showed any changes over time, but its mean value was closer to 1, i.e., highlighting a

comparable mobility inside and outside the site. We unfortunately did not observe enough events of bulging and developing sites to investigate the fluidity at these sites with high precision as compared to Gag accumulation sites showing no change to membrane bulging.

## Discussion

We have developed a smart event-triggered method, denoted etMIN-FLUX, for recording MINFLUX microscopy single-particle tracking data. With etMINFLUX, we extend recent developments in smart microscopy to the state-of-the-art super-resolution capabilities of MINFLUX. The resulting acquisition strategy enables MINFLUX experiments that were previously difficult or impractical, primarily by increasing speed and the throughput of useful trajectories. By triggering MINFLUX acquisitions only at relevant sites and times, etMIN-FLUX ensures that nearly every recorded track contributes meaningfully to downstream analysis and visualization, even though the overall throughput remains intrinsically limited by the single-molecule tracking nature of MINFLUX. In practice, etMINFLUX substantially improves acquisition efficiency. In simpler applications, the temporal overhead time is reduced by approximately a factor of three, while for more complex dynamic processes, the effective throughput of useful data increases by up to sixfold compared with manual acquisition, assuming manual acquisition is feasible at all for the system under study. Moreover, for highly dynamic biological processes, etMINFLUX can reduce phototoxicity by applying MINFLUX illumination only when and where it is required, while the sample is otherwise monitored using low-excitation-power confocal imaging. Beyond MINFLUX itself, we envision that several of the concepts introduced here, such as the ROI-follow acquisition modes that alternate between low- and high-resolution imaging to generate multimodal timelapse datasets, will be broadly useful for other event-triggered microscopy approaches. In this context, the development of a fully generalized framework for event-triggered acquisitions, adaptable to multiple microscope control platforms and software backends, would be highly valuable for the community and could facilitate wider adoption of these methods. Related to this, the etMINFLUX implementation presented here can already be applied to increase throughput and minimize human-in-the-loop control in other imaging scenarios, for example, when performing repeated ROI acquisitions of subcellular structures such as the nuclear pore complex.

The real-time analysis pipelines used here implement different strategies to detect localized intensity changes across both short and long timescales while accounting for their movement. A central challenge for such pipelines is detecting events accurately at an early stage while maintaining high precision. In practice, this introduces a trade-off between recall and precision, since lowering detection thresholds can improve recall but often at the expense of precision, and vice versa (Supplementary Note 6). The performance values observed in the live experiments partly reflect the strong need to optimize pipeline parameters for the specific task, sample preparation, and labeling conditions. In addition, users often prioritize higher recall during optimization, which can reduce precision. Despite these constraints, etMINFLUX achieved robust performance, particularly given the possibility to review detected events either before MINFLUX acquisition or during post-experiment analysis. Further improvements to the pipelines through classical image analysis or machine-learning approaches could potentially enhance both precision and recall. However, such improvements may also introduce trade-offs in computational runtime or detection latency. Increasing pipeline complexity, for example, by adding additional frame-dependent checks, would likely delay event detection and thus reduce temporal responsiveness. Similarly, increasing confocal excitation power could improve detection performance through a higher signal-to-background ratio, but may also increase phototoxicity.

The flexibility of the framework, together with the implemented simulation mode, facilitates the development, testing, and

deployment of upgraded analysis pipelines. Any pipeline that can be implemented as a Python function can be integrated, with each pipeline defined as a separate Python module. This design readily supports machine-learning and deep-learning approaches, provided their inference speed matches the confocal frame rate, as well as GPU-accelerated implementations, as previously demonstrated for etSTED[22]. The framework also allows pipelines to use multiple spectral confocal channels as input. For example, events can be triggered by the dynamic stationarity of an intensity peak together with colocalization of a secondary or tertiary signal, for which we provide example implementations. In addition, pipelines originally developed for etSTED can be adapted with minimal modifications, primarily by adjusting the function call signature. This includes applications such as particle proximity detection. Other readily implementable approaches include the detection of additional dynamical intensity changes, dynamical changes of structural shapes, dynamic feature tracking, or classification-based approaches. Finally, the implementation and hardware allow analysis pipelines to operate either on a per-frame basis or on shorter intervals such as multiple lines. Such options partially alleviate the limitations imposed by the limited confocal acquisition rate used for sample monitoring.

We applied etMINFLUX to three different biological systems and cellular events, which proves its versatility. For more static event sites such as caveolae we highlighted that etMINFLUX can increase the throughput and gather data from hundreds of sites with minimal human interaction; for a rapid event process such as that of budding endosomes we depict that etMINFLUX consistently enables to capture cellular events on time scales down to tens of seconds even in 3D that is unfeasible with any higher throughput with manual control; and for more slowly developing event sites such as Gag accumulation we show that etMINFLUX allows to detect the sites at an early stage and follow the development in 3D at multiple sites simultaneously, enabling any throughput of the data collection. For the nascent Gag-accumulation sites, the reason why not more than 24% of verified event sites show bulging or budding is a question for future studies to further investigate. To our knowledge, this is the first time budding endosomes and viruses have been imaged in 3D and with such high spatial resolution in living cells under physiological conditions, proving the power of the smart etMINFLUX method and topographic data extracted from 3D MINFLUX tracking. Overall, this proves etMINFLUX as a method for expanding the capabilities of MINFLUX to systems where it was previously inapplicable. The implementation framework that controls a commercial abberior MINFLUX system, and is further released open source, will importantly enable a wide range of users to take advantage of the method.

While the applications presented here focus on tracking lipid analogs in membranes for diffusion and topographical studies, the potential application space of etMINFLUX is considerably broader. We envision cellular applications ranging from motor protein tracking to studying molecular exchange in intracellular communication pathways, ultimately limited primarily by current labeling strategies for MINFLUX. etMINFLUX is particularly powerful for investigating fast biological processes with rapid onset on timescales comparable to endocytosis (tens of seconds). In such cases, as demonstrated here, the framework enables MINFLUX data to be acquired with a substantially higher capture rate than achievable through manual microscope control. In addition, the method uniquely enables temporal analysis of slower, minutes-scale cellular processes by allowing multiple events to be precisely aligned to defined biologically relevant time points. Potential applications, therefore, include tracking proteins or small molecules at the plasma membrane, within subcellular compartments, throughout the intracellular space, or along intracellular communication pathways.

## Methods

All research presented here complies with all relevant ethical regulations. No specific ethical permits or board approvals were required.

### MINFLUX microscope

All experiments were performed on a commercial abberior MINFLUX microscope (abberior Instruments GmbH, Göttingen, Germany). Details of the microscope and components are previously fully described[5]. Importantly, the microscope is built around an inverted microscope body (IX83, Olympus, Hachioji, Japan). The lasers used in the presented experiments were: 488 nm (confocal), 642 nm (MINFLUX). In the presented experiments, a stabilization system was used in a feedback loop that stabilizes the sample in all three spatial dimensions. The stabilization system uses a 980 nm IR laser and an xyz sample piezo (Piezoconcept). The objective used was a 100 × 1.45 NA UPLXAPO oil objective (Olympus). Fluorescence detection was performed in two spectral channels for the MINFLUX measurements using two APDs, 650–685 nm and 685–750 nm, from which the signal was summed, and one spectral channel for the confocal measurements using one APD, 500–550 nm. The microscope was controlled using the Imspector control software (v16.3.15635 (m2205) or v16.3.21315 (m2410), abberior Instruments). The microscope was aligned before each measurement session using a gold beads and fluorescent beads sample (abberior Instruments). See summarized information in the Light microscopy reporting table (Supplementary Data 1).

A stage-top incubator with an objective heater (Microscope Heaters, Brighton, UK) was used in order to set a higher temperature in the experiments presented in Fig. 5e–h and Supplementary Fig 22a, c, d, reaching sample temperatures of 34–35 °C. All other presented experiments were performed at room temperature. For the analysis results on the Gag data experiments presented in Fig. 5, a mix between room temperature and incubated experiments were performed. No difference between the two groups was observed. When using an objective heater, the microscope was aligned after the incubator and alignment sample reached a stable high temperature.

### etMINFLUX control widget

The etMINFLUX control widget (further explained in Supplementary Note 1) is a self-standing control widget written in Python, based on the design of the control widget previously developed for event-triggered STED imaging[22] in the ImSwitch environment[60]. The etMINFLUX control widget is entirely self-standing and open source. It is available on GitHub (https://github.com/jonatanalvelid/etMINFLUX) and Zenodo (https://doi.org/10.5281/zenodo.16967460)[61]. The latest version used at the submission of this manuscript is v1.2.0, and its GitHub release includes an executable version. The control widget interfaces with the commercial Imspector control software using the specpy[62] Python package, as well as Python packages mouse[63] and pynput[64] for emulated mouse (minimal and non-critical usage in m2410) and keyboard (only m2205) control, respectively. The control widget has been tested with and versions have been developed for both Imspector version v16.3.15635 (m2205) and v16.3.21315 (m2410), both using specpy v1.2.3 (available with the latest Imspector versions on abberior microscopes) on a commercial abberior MINFLUX setup, running on Windows. The m2410 version is available as the main branch in the GitHub repository, while the m2205 version is available as a separate branch. Other than standard and aforementioned Python packages, the widget has been implemented using the pyqtgraph, PyQt5[65], matplotlib[66], tifffile[67], scipy[68,69], and numpy[70] packages.

### MINFLUX tracking sequences

Custom-adjusted sequences were used for MINFLUX tracking, based on the default fast triangle 2D and octahedron 3D tracking sequences provided by abberior Instruments. The tracking sequences were optimized for each experimental condition and kept consistent throughout all experiments with the same sample type. The most important sequence parameters are listed in Table 1. For all three sequences, the stickiness was set to 4, and the damping was set to 1. Any other parameters were not changed from their default values. The

three sequences are provided as supplementary material. Resulting metadata value distributions (temporal track length, $dt$, and localization brightness, $efo$) are representatively shown for the three different experiment types in Supplementary Fig. 7.

## Confocal and MINFLUX acquisition parameters

Full confocal and MINFLUX acquisition parameters for all data presented in this work are available in Supplementary Table 2. The ranges of confocal and MINFLUX acquisition parameters were the following: 488 nm confocal excitation 0.6–3.0 μW; 642 nm MINFLUX excitation 20–60 μW; MINFLUX ROI sizes 0.6–2 × 0.6–2 μm²; MINFLUX acquisition times 20–90 s; confocal fast axis scanning speeds: 3.5–50 μm/ms; confocal pixel sizes 70–100 nm. See summarized information in the Light microscopy reporting table (Supplementary Data 1).

## Cell culture

HeLa (ATCC CCL-2) cells were cultured in DMEM (9007.1, Carl Roth GmbH + Co. KG, Karlsruhe, Germany) supplemented with 10% (vol/vol) fetal bovine serum (10082147, Gibco, Thermo Fisher Scientific, Waltham, USA) and 1% penicillin-streptomycin (15140122, Gibco, Thermo Fisher Scientific), and maintained at 37 °C and 5% CO2 in a humidified incubator. Cells were seeded on 8-well glass bottom μ-slides (80827 or 80807-90, ibidi GmbH, Gräfelfing, Germany) 24 h (Gag) or 48 h (Dynamin1) before imaging.

PtK2 (ATCC CCL-56) cells were cultured in DMEM (9007.1, Carl Roth) supplemented with 15% (vol/vol) fetal bovine serum (10082147, Gibco, Thermo Fisher Scientific) and 1% penicillin-streptomycin (15140122, Gibco, Thermo Fisher Scientific), and maintained at 37 °C and 5% CO2 in a humidified incubator. Cells were seeded on 8-well glass-bottom μ-slides (80827 or 80807-90, ibidi) 48 h before imaging.

## Transfections

For transfections, cells were transfected 1 day (18–24 h) after seeding using lipid-nanoparticle-based transfection reagent BioT (B01, Bioland Scientific LLC, Paramount, USA) or Lipofectamine 3000 (L3000015, Invitrogen, Thermo Fisher Scientific), according to the instructions of the respective manufacturer. When transfecting with Gag plasmids, both plasmids were used with a ratio of Gag-wt 1:10 Gag-EGFP to ensure a properly functioning Gag lattice formation[71,72]. Transfection times and plasmid amounts were optimized experimentally, qualitatively judging transfection efficiency and cell health with brightfield, widefield, and confocal imaging. Optimized transfection times were 15 h (Dynamin1), 6 h (Gag), and 15 h (Caveolin1). Optimized plasmid amounts were 81 ng/well (Dynamin1), 100 ng/well (Gag), and 125 ng/well (Caveolin1).

## Dye-labeled lipid analogs

Lipid labeling was performed using one of the following dye-labeled lipid analogs: STAR RED-membrane (cholesterol functional group, STRED-0206, abberior GmbH, Göttingen, Germany), STAR RED-C12 sphingosyl PE (SM, STRED-0201, abberior), or STAR RED-1,2-dipalmitoyl-sn-glycero-3-phosphoethanolamine (DPPE, STRED-0200, abberior).

## Caveolae experiments sample labeling

The PtK2 cells were labeled using STAR RED-SM at a concentration of 0.1–0.9 nM or STAR RED-DPPE at a concentration of 0.8–2.5 nM in Leibovitz L-15 medium (21083027, Thermo Fisher Scientific), with the concentration optimized at each experiment day to ensure that the MINFLUX tracking was following single dyes. The cells were immediately after labeling put on the microscope and imaged for a maximum time of 1 h, while ensuring that MINFLUX tracking was following single dyes during the full experiment time.

## Dynamin1 and Gag accumulation sites sample labeling

The HeLa cells were labeled using STAR RED-membrane at a concentration of 150–300 pM for Dynamin1 experiments or 100–300 pM

for Gag experiments, with the dye in L-15, and with the concentration optimized at each experiment day to ensure that the MINFLUX tracking was following single dyes. The cells were incubated with the dye for 5 min at 37 °C, and then washed 2 × with L-15. Fresh L-15 was then added to the wells as imaging medium.

## Plasmids

Cav1-GFP was a gift from Ari Helenius (Addgene plasmid #14433; http://n2t.net/addgene:14433; RRID:Addgene_14433)[73]. pEGFP-N1 Dynamin1 wt was a gift from Justin Taraska (Addgene plasmid #120313; http://n2t.net/addgene:120313; RRID:Addgene_120313)[74]. Gag-EGFP and Gag wt were gifts from Jennifer Lippincott-Schwartz[75].

## Gold beads labeling

For use with the XYZ sample stabilization system of the MINFLUX microscope, 150 nm gold colloid particles (EM.GC150/4, BBI Solutions, Crumlin, United Kingdom) were added to each sample. At the beginning of the sample preparation protocol, ~30 min before experiment start and ~15 min before lipid dye labeling, a solution of 120 μl stock solution in 400–450 μl L15 imaging medium was added to the cells. The gold bead solution was left to incubate for 15 min at 37 °C, followed by washing 6–8 × with 250 μl L15 imaging medium.

## Post-acquisition data analysis

All data analysis and data visualization performed to produce the results in this work has been performed with custom Python-based analysis pipelines. A public GitHub repository is available with all the scripts necessary to analyse the data and produce the visualizations: https://github.com/jonatanalvelid/etMINFLUX-analysis-public. The data saving structure and suggested data handling routines are further described in Supplementary Note 13.

## Confocal-MINFLUX scan shift compensation

Coordinate system scan shifts between confocal and MINFLUX data were observed and compensated for according to necessity in post-processing of the acquired data. This was applied for all Caveolin1 event data. The procedure for measuring and applying shift compensation is further explained in Supplementary Note 3.

## Diffusion and transient diffusion analysis

Diffusion analysis was performed using square displacement analysis and subsequent fitting of the population of displacements and time steps with a function representing a free Brownian motion in two dimensions, i.e., on the membrane. A 2D Brownian diffusion model in a limited time step span, using a maximum time lag of 500 μs or 5 ms for the 2D and 3D MINFLUX tracking diffusion analysis, respectively, was always assumed as we analyzed the short-range diffusion of lipid analogs in the membrane to investigate the immediate diffusion at sites never larger than a few hundred nanometers. From each fit, the diffusion coefficient and dynamic localization precision could be extracted. Diffusion analysis was performed on segments of tracks, while transient diffusion analysis was performed for each localization using a sliding window centered on the localization. For further details of the diffusion analysis, see Supplementary Note 9.

## Packing coefficient analysis for hop diffusion analysis

Packing coefficient analysis was performed following the method described by Renner et al[39]. Briefly, the spatial extent of a track segment centered on each localization was evaluated using a sliding-window approach. The packing coefficient was defined as the sum of the squared distances between successive localizations divided by the square of the convex hull area enclosing the segment. In this metric, high packing coefficient values indicate stronger spatial confinement. The analysis was performed using a packing coefficient threshold of 880, corresponding to an equivalent confinement diameter of 70 nm,

together with a temporal confinement window threshold of 10 ms, to avoid noisy detections, and a sliding window of 25 localizations. Only tracks containing at least 100 localizations were included. A segment with a packing coefficient above the threshold was classified as confined only if it persisted for at least 10 ms and contained at least 25 localizations. Undefined packing coefficient values were replaced with the value of the nearest valid neighbor. From the analysis, we extracted the number of confinement events per track. For tracks exhibiting confinement, we further calculated the fraction of time spent in the confined state, the number of confinement events per unit time, and the radial dependence of confinement, expressed as the temporal fraction of confinement as a function of the distance between each localization and the center of the event site.

### Caveolae sites analysis results filtering

Thanks to the relatively large amount of data recorded, the final analysis results considered for the caveolae sites were filtered according to the following considerations in order to ensure robust results. For diffusion coefficient analysis, only those track segments where the number of square distances was above 15 were considered, and only those sites where the number of fitted diffusion coefficients, i.e., the number of track segments, inside and outside were both above 5 were considered. Furthermore, the mean diffusion coefficient of the track segments inside and outside were both required to be above 0. For the transient diffusion coefficients, only the sites where the total number of tracks was above 20 and where the number of localizations inside was above 50 were considered. For the inclusion ratio T, only ROIs recorded with the same size, $1 \times 1 \, \mu m^2$, and where the total number of tracks was above 20 were considered.

### Data visualization

*3D surface fitting*. For 3D surface visualization of MINFLUX tracking data, we use the ch-shrinkwrap[76] localization cloud surface fitting plugin to PYME[24]. To generate a shrinkwrap surface, an isosurface is first fitted to the localization cloud. This is performed with the following parameter value ranges: N points min, 12–20; threshold density, 0.00001–0.0001; inner surface culling, remeshing, and smooth curvature were used. Following this, the shrinkwrap algorithm is performed on the created isosurface, with the following parameter values or value ranges: curvature weight, 7; remesh frequency, 5; neck first iteration, 5; neck threshold low, − 0.00003–−0.001; neck threshold high, 0.01; punch frequency, 0; finishing iters, 0; Kc, 0.9; minimum edge length, 5.0; truncate at, 1000; smooth curvature was used. *3D point cloud z-threshold*. In the following 3D point cloud visualizations, as well as overlay between confocal and a 2D projection of the localization point cloud, a z-threshold has been used to only extract the data from the membrane on which the event of interest is occurring: Fig. 5g, h, and Supplementary Fig 22a. On all of these occasions, both cellular membranes have been captured and are present in the raw 3D MINFLUX tracking, and the threshold has been introduced manually for each case in order to allow for accurate analysis and clearer visualization. *Supplementary movies*. All supplementary movies were created in napari using the plugin napari-animation together with our plugin napari-etminflux-data-viewer (Supplementary Note 4).

### Localization precision determination

The localization precision was chosen to be extracted as the dynamic localization precision from fitting of the square displacement analysis, rather than subsequent step distance analysis, due to the diffusional nature of the tracked molecules. With the linear fitting model used (Supplementary Note 9), the diffusion coefficient can be extracted as the slope, while the dynamic localization precision can be extracted as the intersection with the vertical axis. All tracks were fitted individually, the dynamic localization precision was extracted from each track where the number of square distances from the track was above 15, and

the individual datapoints in Supplementary Fig. 12 represent the mean of all tracks in a region of interest around a detected event site. For the 2D dynamic localization precisions from the experiments at caveolae sites, this means that each data point represents one region of interest, while for the 3D dynamic localization precisions from the experiments on virus budding sites, it means that each data point represents one cycle recorded at one region of interest.

### Statistics

Statistical tests are either one-sample $t$ tests for the null hypothesis that the mean is equal to 1 (Fig. 3g, h), or two-sample $t$ tests for the null hypothesis that the two independent (Fig. 3i, j) or related (Supplementary Fig. 13) populations have identical means. Statistical significance is presented in the following way: n.s., $p > 0.05$; *, $p < 0.05$; **, $p < 0.01$; ***, $p < 0.001$.

### Experimental sizes

Total number of experiment days: 5/5/3/3 (Caveolin1: SM-Cav1/DPPE-Cav1/SM-random/DPPE-random), 4 (Dynamin1), 12 (Gag). Total number of samples: 13/14/5/5 (Caveolin1: SM-Cav1/DPPE-Cav1/SM-random/DPPE-random), 10 (Dynamin1), 35 (Gag). Total number of cells: 17/30/7/7 (Caveolin1: SM-Cav1/DPPE-Cav1/SM-random/DPPE-random), 33 (Dynamin1), 42 (Gag). Total number of detected events: 101/141/31/29 (Caveolin1: SM-Cav1/DPPE-Cav1/SM-random/DPPE-random), 155 (Dynamin1), 106 (Gag).

### Reporting summary

Further information on research design is available in the Nature Portfolio Reporting Summary linked to this article.

## Data availability

The confocal imaging, MINFLUX tracking data, log files, and metadata generated and used in this study, supporting the method implementation and the findings, have been deposited in Zenodo with reference number 15608839 https://doi.org/10.5281/zenodo.15608839[77]. Numerical data underlying graphs as presented in this article are provided in the Source Data file. Source data are provided in this paper.

## Code availability

The source code of the developed control widget in this study is open source and available on GitHub: https://github.com/jonatanalvelid/etMINFLUX and in a released version, v1.2.0, at Zenodo, reference number 16967460: https://doi.org/10.5281/zenodo.16967460[61]. etMINFLUX v1.0.0 was used for the majority of data collection in this manuscript. The software is also available as an executable in the v1.2.0 release in the GitHub repository. The code of the analysis supporting the findings in this work, in the forms of Jupyter notebooks, Python scripts, and Python class modules, together with example data, is available on GitHub: https://github.com/jonatanalvelid/etMINFLUX-analysis-public and Zenodo, reference number 16967493: https://doi.org/10.5281/zenodo.16967493[78]. The data viewer napari plugin is available on GitHub: https://github.com/jonatanalvelid/napari-etminflux-data-viewer, directly from PyPI: https://pypi.org/project/napari-etminflux-data-viewer/, and in a released v1.0.2, on Zenodo, reference number 18712437: https://doi.org/10.5281/zenodo.18712437[79].

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

## Acknowledgements

The authors thank Annett Urbanek for help with sample preparation and the Microverse Imaging Center (and Aurélie Jost / Patrick Then) for providing general microscope facility support for data acquisition (and data analysis).

## Author contributions

C.E. supervised the research. C.E. and J.A. funded the research. J.A. and C.E. conceived the project idea. J.A. and C.E. designed the research. J.A. planned and developed the control widget and the real-time analysis pipelines; planned, prepared, and performed experiments; and performed data analysis. A.K. planned, prepared, and performed experiments and provided experimental and data analysis guidance. J.A. drafted the manuscript with input from all the authors. All authors revised and accepted the final version of the manuscript.

## Funding

J.A. discloses support for the research of this work from the Swedish Research Council [2022-06139_VR]. The authors discloses support for the research of this work from the German Research Foundation (DFG) [Instrument funding MINFLUX Jena INST 275_405_1, Germany's Excellence Strategy – EXC 2051 – 390713860, SFB 1278 – 316213987, GRK M-M-M: GRK 2723/1 – 2023 – 44711651, GRK PhInt: GRK 3014/1, instrument funding 460889961 multi-photon laser scanning device], the Leibniz Association [Leibniz Collaborative Excellence Program, AMPel – K548/2023], the Free State of Thuringia [TAB, Advanced Flu-Spec / 2020 FGZ: FGI 0031, Multi-XUV / 2023 FGR 0054, Thimedop 2018 IZN 0002], the Federal Ministry of Research, Technology and Space [LIVE2QMIC, FGZ: 13N15956], the Photonics Research Germany [FKZ: 13N15713 / 13N15717], the Leibniz Center for Photonics in Infection Research (LPI) [BMFTR roadmap], and the European Union within the framework of the European Regional Development Fund (EFRE). Open Access funding enabled and organized by Projekt DEAL.

## Competing interests

The authors declare no competing interests.
