## [Transparent Peer Review file · Nature Communications]

Smart event-triggered MINFLUX microscopy to catch and follow rare events

Corresponding Author: Dr Jonatan Alvelid

Version 0:

Reviewer comments:

Reviewer #1

(Remarks to the Author)

MINFLUX microscopy has been an established state-of-the-art method for ultra-high precision single-molecule imaging, but the applications to rare events remain challenging due to limited field-of-view and laborious demand on human operator. This study presents an innovative strategy for automating MINFLUX across scales. Using a commercial MINFLUX system as the hardware platform, the authors develop a number of open-source software modules that make use of broader field confocal imaging for coarse monitoring of desirable events. When the criteria are met, MINFLUX acquisition in the automatically defined subregions are activated. Several modes of operations offered, tailored to specific biological contexts, which are primarily focused on lipid dynamics in this study.

In the first example, Cavelolin-GFP signal was monitored in confocal mode, and MINFLUX acquisition activated to monitor the trajectories of DPPE and sphingomyelin. A small reduction in diffusion coefficients of sphingomyelin is observed, whereas DPPE shows no difference. In the second example, Dynamin1-GFP was used to mark endocytic budding sites, with 3D MINFLUX of cholesterol-based membrane probe performed after event detection. From the analysis of the 3D trajectories, this enable the reconstruction of the endosome topography as defined by diameter and neck length. In the third example, budding of Gag-labeled virus like particular was performed, enabling live visualization of VLP budding. Overall, this is an important and timely investigation that aims to make MINFLUX more broadly accessible and useful for the cell biology community. However, there are a number of technical presentation issues that I recommend attending to as described further below. The discussion is also very short and does not provide much critical analysis of limitations or further possible improvements, nor do the findings made here put into the context of previous literature. For example, while the event-detection scheme used here appear to significantly increase the number of detected events compared to manual acquisition, there are still examples such as Gag where the authors mentioned not being able to detect rare events in sufficient numbers. Trajectories number in the hundreds, whereas in conventional SPT, the trajectory numbers can be orders of magnitude higher. Is this an intrinsic or practical limitation? More generally, as the main innovation of this study is primarily in the software control aspect, and not in hardware development (using turnkey commercial system in this case), one would expect a more extensive demonstration of biological applications, ideally leading to some biological findings. For example, in the current version, the cells are imaged only in the unperturbed state. The authors may wish to demonstrate an example of how various pharmacological or genetic perturbations affect the dynamics or morphological features of these membrane structures. This will help provide a concrete instructive example on how a biologist may be using this method for.

Major comments :

1. Event-detection is obviously the key to this study but the description of what constitutes 'event' and their accuracy are not clearly included in the main manuscript. One would expect this to be more clearly mentioned in the first section of the results, but this section in its current form reads more like a software manual than a research manuscript. The authors discuss event detection separately for each example later on, with most of the details as text in the supplementary materials, but it seems better articulation and illustration would be highly beneficial to the reader. One would expect a systematic investigation of different event-detection modes and evaluations of false positives/negatives. This is particularly important when the stated aim of the study is disseminating this software to the community, since the large amount of raw data involved would make it very challenging for end users to validate event detection accuracy. Thus, the onus is on the developer to rigorously establish the validity of this, and have a clear workflow delineated.
2. The important parameters for interpreting the trajectories are the localization precision, and the time-step (equiv. to frame-rate). These are presented collectively in supplementary figures. The localization precision is reported only for STAR RED even though ATTO647N is also used. As much as possible, these parameters should be included in the main figures when

single-molecule trajectories are shown.

3. From visual inspection of the trajectories such as in Fig. 3d-e, it would seem that the trajectories could exhibit segment-specific diffusion behavior, perhaps hopped diffusion behavior. Is this the case, and are there differences between regions inside or outside caveolae?

4. Diffusion dynamics of sphingomyelins have been studied extensively in the literature, and there are various perturbations that affect its mobility. The authors only show a comparison between SM and DPPE, with the magnitude of the difference seemingly quite subtle compared to what has been reported by different imaging modality. As this should be a proof-of-principle study, it would be helpful for the authors to investigate SM mobility under a broader range of common perturbations used in the literature (e.g. cholesterol etc).

5. In Supplementary Fig. 1c-d, the traces seem to show irregular patterns and appear to be composed of three distinct subpopulations rather than a single one. What is the origin of this?

6. Supplementary Figure 4 is difficult to interpret. Furthermore, it reports a 3D localization precision that is superior to the 2D precision, which is counterintuitive based on the specification of the commercial MINFLUX. Further clarification would be helpful.

(Remarks on code availability)

Reviewer #2

(Remarks to the Author)

The authors present the application of the event-triggered acquisition concept to MINFLUX tracking in 2D and 3D. The framework uses large confocal images to detect localized events of interest and triggers high resolution MINFLUX tracking only when and where necessary. This allows for improved throughput and the study of otherwise hard to capture biological events. Both single and multi/ROI imaging are possible using a GUI-supported open-source Python software that interacts with the commercial Inspector software via the `specpy` package and with keyboard and mouse emulation.

The performance of the framework is exemplified with three applications in the membrane diffusion space: lipid diffusion at caveolae, budding endosomes, and membrane fluidity at Gag accumulation sites. Intensity-based image analysis pipelines observe large areas and trigger localized MINFLUX tracking procedures. The authors perform diffusion analysis to compare different areas in the membrane and fit surfaces to the observed tracks in order to visualize the membrane structure.

While etMINFLUX shows promise for studying rare cellular events with unprecedented detail, better communication about the limitations of the method and its implementation would strengthen the manuscript considerably.

Major Concerns

The etMINFLUX concept is well introduced with clear mode descriptions. However, the connection to etSTED is largely lost in the main manuscript and not discussed deeply enough in the Supplementary Material. Looking at the Controller code, the two packages share many concepts. The authors should better explain which aspects of event-triggered imaging were developed specifically for etMINFLUX versus adapted from etSTED. Why was the etSTED-widget-base not used with a different Inspector backend? The ROI follow modes seem like a general concept that could be used for other smart microscopy applications. Generalizing here could strengthen the manuscript's impact outside the MINFLUX space.

For a comprehensive understanding of etMINFLUX benefits, I am missing direct quantitative comparisons of data acquisition with and without event-triggered acquisitions. How long would it have taken to acquire the data conventionally? Would some results not have been possible at all, and why? Where are the bottlenecks? MINFLUX is especially beneficial due to low excitation intensities—how does confocal imaging influence phototoxicity in these experiments?

The limitations for adapting to different biological applications should be better discussed. Mainly in two areas:

The analysis pipelines rely on intensity changes and seem similar to one another. What other approaches would be readily supported? Both by the framework and the hardware (e.g. confocal colocalization analysis, shape analysis, dynamics, neural networks).

The presented applications are limited to membrane diffusion. What are the limitations to applying this to other biological problems e.g. motor tracking in 3D?

As a reader from the technical side, I have difficulty understanding the impact of the biological findings. Some diffusion detail could be shifted to Supplementary Material, allowing the main manuscript to provide higher-level biological context with explanations for technical readers regarding the data's impact.

Minor Concerns

The authors state in the discussion that application to MINFLUX imaging would be straightforward. While I agree technically, applications of smart microscopy for fixed samples are typically harder to identify and dynamic imaging is hard using MINFLUX. I also see the combined complexity of MINFLUX imaging plus an event-triggered framework as being greater than for tracking, making this adaptation not as trivial as it might seem at first.

I can't make much sense of the denser diffusion plots (e.g., Fig. 4f)

It would be great to have compiled dynamic data as videos to get a better feeling of the dynamics of the experiments. Or/and even a napari plugin or script to load the confocal data together with the tracks.

The performance values for the analysis pipelines are relatively low. The authors should discuss the impact and type of false positives and how the performance of these pipelines could be further improved.

(Remarks on code availability)

Control

The repository is well documented and the code is well structured. I was not able to test the software, as newer `specpy` versions are only available with `lmspector`. I tried to get the simulation environment working under Python 3.6 with the publicly available `specpy` 1.2.1, which was not possible due to several compatibility issues. I think the authors should comment more extensively on how they view this situation, especially given that the public `specpy` version was last updated in 2017. Is there a commitment from Abberior to maintain compatibility in the future? Are there alternative ways to run the software if newer or updated systems no longer support `specpy`?

The use of emulated mouse control seems fragile to me. The authors mention in the Supplementary Material that an extension of the API would solve this issue. Are there efforts underway to implement such an extension? Otherwise, I assume the system cannot be touched during the experiment to avoid interfering with the simulated mouse control. In this case, the authors should temper the parts in the manuscript that mention on-the-fly adjustments to imaging parameters.

While the documentation of the setup and calibration steps is extensive and understandable, I see several problematic areas that need improvement to ensure reusability. The mention of specific line numbers in the code is prone to becoming outdated with code updates. I think these settings should be extracted to external configuration files with clearly structured and named fields. The calibration procedure for pixels might not work for zoomed views, and new users will need considerable expertise to optimize timing settings. A more streamlined process would ensure better reusability by testing timing on a specific system and suggesting optimized values for example.

For users without direct access to a MINFLUX system (for example, when using it at a core facility), it would be important to have a simulation environment to test analysis workflows in the context of `etMINFLUX` on existing data. Given the problematic dependencies, this seems non-trivial, even with the mocked `lmspector` instance.

Analysis

The analysis code is structured well and data flow is well documented.

Reviewer #3

(Remarks to the Author)

The manuscript "Event-triggered MINFLUX microscopy: smart microscopy to catch and follow rare events" by Alvelid et al. addresses a highly relevant and timely issue namely how to bridge the very limited spatial throughput of the new and powerful microscopy modality MINFLUX with imaging in living cells where everything moves and relevant things can happen and appear at unknown positions in both space and time. To resolve this conundrum the authors have developed an event-triggered imaging approach termed `etMINFLUX`. With this approach the authors either image a single large field-of-view (FOV) and algorithmically identify multiple sites of interest and do targeted MINFLUX imaging at these sites. Alternatively, the authors let the automated control software monitor a large FOV at regular intervals with standard confocal microscopy and subsequently upon identification of an "event" corresponding to a predefined criteria do targeted MINFLUX imaging within a small FOV around the identified event. The authors apply the method to three different biological systems: 1, the diffusion of lipids through caveolae; 2, endosomal budding from the plasma membrane; and 3, lipid diffusion and shape determination of Gag-protein positive HIV-1 (human immunodeficiency virus) budding sites. The work is clearly presented and highly relevant. Indeed, from my perspective, the presented approach seems to be basically a necessary add-on to most real-world scientific questions one would want to address with the high resolution MINFLUX technique given this technique's limited FOV. The obtained results speak to the usefulness of the approach as it would have been difficult to obtain such large numbers of observations as reported (of the three different examples) without the use of the automated strategy presented. There are a few limitations and minor issues presented below but in general the validity and usefulness of the approach is very sound, both from a theoretical and a practically achieved perspective.

Minor issues:

- 1) In figure 3 the magnitude of the differences between the diffusion properties of both the two studied lipids (SM and DPPE) and within and outside of caveolae appear quite small. This is not a critique of the usefulness of the method or the validity of the results but as an example of effects on lipid diffusivity it is perhaps not very striking, which to some extent limits the interpretation of what sensitivity/dynamic range of measurable diffusion changes within structures can be measured with this approach.
- 2) In figure 4f and g, example membrane topology maps and their relative frequency at dynamin1 sites are presented. To get a better appreciation of the specificity of the endosomal invaginations it would have been valuable to see a similar analysis done on random non-dynamin1 positive sites.
- 3) Similar to issue #2 above, in figure 5g-l it would have been very informative to see a similar analysis of random membrane sites.

(Remarks on code availability)

The code and documentation appear well organized and possible to follow to install and use. I have not reviewed the code in detail as this is beyond my expertise.

Version 1:

Reviewer comments:

Reviewer #1

(Remarks to the Author)

I thank the authors for comprehensively revised the manuscript. The revision has largely addressed the comments I raised for the original version while also greatly improve the readability. I now support its acceptance.

(Remarks on code availability)

Reviewer #2

(Remarks to the Author)

The authors have addressed all concerns from my initial review. The additions of quantitative throughput comparisons, the napari visualization plugin, and simulation mode significantly strengthen the manuscript's practical impact. The expanded Discussion appropriately contextualizes the work within smart microscopy developments and discusses limitations more clearly. The revised implementation using the updated Inspector API eliminates the fragile mouse emulation, and the relationship to etSTED is now well explained.

The authors have strengthened both the technical presentation and practical accessibility of the work. The combination of rigorous benchmarking, improved software tooling, and comprehensive documentation makes this a solid contribution to the smart microscopy field.

(Remarks on code availability)

I was now able to install all necessary packages for the simulation environment and launch the widget. The readme covers all important aspects.

Reviewer #3

(Remarks to the Author)

My concerns have been addressed appropriately.

(Remarks on code availability)

Dear Dr Alvelid,

Thank you again for submitting your manuscript "Event-triggered MINFLUX microscopy: smart microscopy to catch and follow rare events" to Nature Communications. We have now received reports from 3 reviewers and, based on their comments, we have decided to invite a revision of your work. Your revision should address all the points raised by our reviewers (see their reports below).

When resubmitting, **you must provide a point-by-point response to the reviewers' comments. Please show all changes in the manuscript text file with track changes or colour highlighting.** If you are unable to address specific reviewer requests or find any points invalid, please explain why in the point-by-point response.

We hope to receive your revised paper within three months, but we understand that revisions may take longer. Please let us know if you find that the revision process will take substantially more time.

When evaluating your revised manuscript, we will not consider any similar papers published independently in the meantime to compromise the novelty of your study. See here for more information.

Best regards,

Nelio Rodrigues, PhD
Senior Editor
Nature Communications

Dear Editor and Reviewers,

Thank you for your careful review of our manuscript. We are pleased to submit a revised version together with a detailed point-by-point response to all reviewer comments below. We believe that the revisions and clarifications have substantially improved the manuscript, and we hope that the changes address the reviewers' concerns satisfactorily. For clarity, all questions have been numbered (R[reviewer#]Q[question#]) and our corresponding responses are labelled (R[reviewer#]Q[question#]R). **Reviewer comments are shown in black and our replies in blue.** All references to figures, sections, and line numbers correspond to the numbering in the revised manuscript (line numbers correspond to the numbering in the version with tracked changes).

REVIEWER COMMENTS

Reviewer #1 (Remarks to the Author):

MINFLUX microscopy has been an established state-of-the-art method for ultra-high precision single-molecule imaging, but the applications to rare events remain challenging due to limited field-of-view and laborious demand on human operator. This study presents an innovative strategy for automating MINFLUX across scales. Using a commercial MINFLUX system as the hardware platform, the authors develop a number of open-source software modules that make use of broader field confocal imaging for coarse monitoring of desirable events. When the criteria are met, MINFLUX acquisition in the automatically defined subregions are activated.

Several modes of operations offered, tailored to specific biological contexts, which are primarily focused on lipid dynamics in this study.

In the first example, Cavelolin-GFP signal was monitored in confocal mode, and MINFLUX acquisition activated to monitor the trajectories of DPPE and sphingomyelin. A small reduction in diffusion coefficients of sphingomyelin is observed, whereas DPPE shows no difference. In the second example, Dynamin1-GFP was used to mark endocytic budding sites, with 3D MINFLUX of cholesterol-based membrane probe performed after event detection. From the analysis of the 3D trajectories, this enable the reconstruction of the endosome topography as defined by diameter and neck length. In the third example, budding of Gag-labeled virus like particular was performed, enabling live visualization of VLP budding.

Overall, this is an important and timely investigation that aims to make MINFLUX more broadly accessible and useful for the cell biology community. However, there are a number of technical presentation issues that I recommend attending to as described further below. The discussion is also very short and does not provide much critical analysis of limitations or further possible improvements, nor do the findings made here put into the context of previous literature. For example, while the event-detection scheme used here appear to significantly increase the number of detected events compared to manual acquisition, there are still examples such as Gag where the authors mentioned not being able to detect rare events in sufficient numbers. Trajectories number in the hundreds, whereas in conventional SPT, the trajectory numbers can be orders of magnitude higher. Is this an intrinsic or practical limitation? More generally, as the main innovation of this study is primarily in the software control aspect, and not in hardware development (using turnkey commercial system in this case), one would expect a more extensive demonstration of biological applications, ideally leading to some biological findings. For example, in the current version, the cells are imaged only in the unperturbed state. The authors may wish to demonstrate an example of how various pharmacological or genetic perturbations affect the dynamics or morphological features of these membrane structures. This will help provide a concrete instructive example on how a biologist may be using this method for.

R1NR1. We thank the reviewer for his/her thoughtful overall assessment. We address the specific points in detail below, but would first like to summarize several major revisions made to the manuscript in response to these comments.

The Discussion section has been substantially expanded to better place the work in context, outline current limitations, and discuss potential future developments. In addition, for the Dynamin1 and Gag experiments we have added direct comparisons to manually controlled acquisitions, allowing us to quantify the improvements in throughput achieved with the automated framework.

Regarding the comment on the relatively low number of Gag events detected, we would like to reiterate that what we do comment on is that for the diffusion analysis, the number of events showing a bulging or budding are not enough to reach a high enough statistical certainty in comparing these sites to sites that show a flat membrane. It is important to note that sites that do show a flat membrane are still valid event detections in that they show an increasing Gag signal before and after event detection. However, the reasons why more Gag-accumulation sites do not show a bulging and eventual budding is something we are unsure of. A possibility is that there is biological relevance in that all Gag-accumulation sites do not form a proper Gag lattice that can induce the membrane bending we are looking for, which future studies have to investigate. As mentioned further down in this reply, we are currently performing a study using etMINFLUX to further study virus budding sites, and in the realm of those experiments, when entering already-present accumulated Gag sites, bulging and budding Gag+ sites in various stages are more commonly found in the 3D MINFLUX data. We have

clarified the statement about the diffusion analysis, and further added a comment in the Discussion regarding the low amount of bulging/budding in Gag+ sites and that future studies have to further investigate this.

Regarding the comparatively low trajectory numbers relative to conventional single-particle tracking (e.g. camera-based widefield SPT), this is an intrinsic limitation of MINFLUX rather than of the etMINFLUX approach presented here. Because MINFLUX tracks individual fluorophores sequentially, overall throughput is inherently lower than in widefield SPT. One of the key motivations for etMINFLUX is to mitigate this limitation by triggering acquisitions precisely when and where relevant events occur, thereby maximizing the fraction of trajectories that are informative for downstream analysis. We now discuss this point explicitly in the early Results section and in the expanded Discussion. Also, the big advantage of MINFLUX is its ability to reveal three-dimensional structures in conjunction with diffusion values in such a precise way, impossible with conventional widefield or TIRF SPT.

With respect to the suggestion of further biological demonstrations, we would like to emphasize that the primary goal of this study is to present and validate a novel method and methodological framework implementation of that method. While no new hardware is introduced, the software development is substantial and enables fundamentally new experimental workflows on a commercial MINFLUX platform, allowing experiments that are severely limited under manual control. We have clarified this methodological focus in the revised manuscript.

Extensive biological studies using this framework are currently ongoing. For example, we are applying the approach to investigate lipid composition and dynamics at viral budding sites and have developed a multicolour event-detection pipeline to study peroxisomal membrane proteins, which are samples where manual acquisition severely limits throughput. These projects are aimed at addressing specific biological questions and will be presented in dedicated manuscripts. See the response in R2Q4R for further discussion on this topic, where also Rebuttal Figure 1 shows an example etMINFLUX experiment from our current study on peroxisomal membrane protein tracking. Including such studies here would shift the focus away from the methodological advances of the present work. Instead, the manuscript is intended to introduce the method, introduce our practical framework, and demonstrate their capabilities across several biological examples of increasing complexity. To further strengthen this technical perspective, we have added comparisons with manual and signal-negative-triggered acquisitions to quantitatively illustrate the performance gains provided by the automated framework. Together with the expanded discussion of methodological implications and future developments, we believe these revisions significantly strengthen the manuscript and clarify its scope.

Major comments :

R1Q1. Event-detection is obviously the key to this study but the description of what constitutes 'event' (1) and their accuracy (5) are not clearly included in the main manuscript. One would expect this to be more clearly mentioned in the first section of the results (1), but this section in its current form reads more like a software manual than a research manuscript (2). The authors discuss event detection separately for each example later on, with most of the details as text in the supplementary materials, but it seems better articulation and illustration (3) would be highly beneficial to the reader. One would expect a systematic investigation of different event-detection modes and evaluations of false positives/negatives (4). This is particularly important when the stated aim of the study is disseminating this software to the community, since the large amount of raw data involved would make it very challenging for end users to

validate event detection accuracy (5) Thus, the onus is on the developer to rigorously establish the validity of this, and have a clear workflow delineated (6).

R1Q1R. Thank you for these helpful comments. In response, we have revised the manuscript extensively by adding clarifications, figures, experiments, and analyses to better explain the method and framework. Below we address the points raised by the reviewer.

1. Definition of an event. We now introduce a general definition of an “event” in the Introduction (lines 51-55). This provides a conceptual framework for event-triggered microscopy, while the specific event definitions for the three applications are described in detail in the Results. To further clarify this, the final paragraph of the Introduction now briefly defines the events used in each application (lines 69-77). The expanded Discussion also highlights potential future event types and application areas.

2. Presentation of the Results section. We have revised the opening part of the Results to make it less manual-like and more focused on the scientific rationale and key methodological concepts. Technical implementation details of the Python framework and microscope interface have been shortened and partly moved to the Supplementary Notes (mainly Supplementary Note 1 and Supplementary Note 3).

3. Illustration of event detection. To better illustrate the event detection types, we added a new Supplementary Fig. 3. This figure visually summarizes each analysis pipeline and consequently the conditions and thresholds required to trigger an event for each application, together with representative examples. Combined with the general definition introduced in the main text, this should clarify both the specific implementations used here and the broader applicability of the framework.

4. Evaluation of event detection performance. We assume the reviewer refers to the analysis pipelines used for event detection (rather than the acquisition modes such as single-ROI or multi-ROI operation). Event detection accuracy depends primarily on the analysis pipeline and the biological sample rather than on the acquisition mode itself. In the original manuscript we already reported true/false event detection statistics for the Dynamin1 and Gag pipelines (Fig. 4h and 5i). We have now extended this analysis by performing additional manual annotations and automated Bayesian parameter optimization. The optimized pipelines were evaluated on pre-recorded confocal time-lapse datasets, and the results are presented in a new Supplementary Fig. 11 with details in Supplementary Note 6, and summarized in respective Results section (lines 402-405, lines 517-523). We report precision, recall, and weighted F_β scores for each dataset. The weighting was chosen according to the biological context, for example, prioritizing precision for relatively frequent events (Caveolin1 and Dynamin1) and recall for rare events (Gag). It is important to note that these metrics depend on the specific labelling, imaging conditions, and sample preparation used here. Therefore, they should be interpreted as benchmarks demonstrating the achievable performance under similar conditions rather than universally transferable values.

5. Applicability and validation by end users. We agree that careful documentation and validation are essential when releasing such a framework. At the same time, event detection performance inevitably depends on experimental conditions (e.g., labelling strategy, fluorophores, expression levels, imaging parameters). While we can rigorously evaluate the pipelines for the conditions used in this study (as now described above), users will typically need to optimize parameters for their own samples. The framework was therefore designed to allow straightforward adjustment of analysis parameters through the graphical interface. To facilitate validation, the software saves confocal data, MINIFLUX data, and metadata in a structured format that enables systematic inspection of detected events. We also provide analysis scripts that allow users to evaluate event detection performance for datasets similar

to the Dynamin1 and Gag examples. The flexibility of the framework also allows users to implement new event-detection pipelines tailored to other biological questions, all while still providing access to the various recording modes and technicalities of the framework. In ongoing work, for example, we are extending the approach to detect events of temporally stationary structures with dual-colour labelling. Such use cases will require different validation procedures, which is why the framework focuses on providing flexible tools and clear data structures rather than a single universal validation pipeline.

6. Data workflow and organization. To further clarify data handling, we added a detailed description of the data structure and workflow in a new Supplementary Note 13. This section outlines how experimental data is stored, how events can be inspected and curated, and how analysis scripts can be used for visualization and quantitative analysis. In addition, as described in more detail in our response to Reviewer 2 (R2Q8R), we implemented a new *napari* widget that enables interactive visualization of etMINFLUX datasets, including combined views of confocal images, MINFLUX trajectories, and associated metadata, from any recording mode of the framework. We believe these additions provide sufficient guidance for users to interpret and analyse their datasets using the framework.

R1Q2. The important parameters for interpreting the trajectories are the localization precision, and the time-step (equiv. to frame-rate). These are presented collectively in supplementary figures. The localization precision is reported only for STAR RED even though ATTO647N is also used. As much as possible, these parameters should be included in the main figures when single-molecule trajectories are shown.

R1Q2R. Thank you for this comment. We agree that localization precision and time step are critical parameters for interpreting MINFLUX trajectories. We have therefore added corresponding plots to the main Figs. 3–5 and Extended Data Figs. 4–8, showing the distributions of localization precision and time steps (per dataset / ROI cycle), together with track length distributions for each dataset. These are briefly summarized in the main text (lines 242-244, lines 387-389, lines 497-499).

Regarding the fluorophores used, we exclusively employ STAR RED for MINFLUX tracking throughout the study, as stated in the manuscript. ATTO647N was not used in any of the experiments. We carefully checked the main text, figures, and supplementary materials but could not identify any mentions of ATTO647N. It is therefore possible that this comment was based on a misunderstanding; however, we would be happy to correct any remaining inconsistencies if the reviewer can indicate where ATTO647N may have been mentioned inadvertently.

R1Q3. From visual inspection of the trajectories such as in Fig. 3d-e, it would seem that the trajectories could exhibit segment-specific diffusion behavior, perhaps hopped diffusion behavior. Is this the case, and are there differences between regions inside or outside caveolae?

R1Q3R. Thank you for this interesting suggestion. To investigate the possibility of segment-specific or hop-like diffusion behaviour, we have added a new analysis of lipid analogue diffusion at and around Caveolin1-positive sites.

Specifically, we have performed packing coefficient analysis as described by Renner et al. (2017)¹, which enables the detection of transient spatial confinements within single-particle tracks. The method evaluates the spatial extent of track segments within a sliding window

¹ Marianne Renner et al., “A Simple and Powerful Analysis of Lateral Subdiffusion Using Single Particle Tracking,” *Biophysical Journal* 113, no. 11 (2017): 2452–63, <https://doi.org/10.1016/j.bpj.2017.09.017>.

around each localization, where the packing coefficient is defined as the sum of squared displacements divided by the squared convex hull area of the segment. High packing coefficient values indicate spatial confinement. Using this approach, we identified confined track segments by applying a packing coefficient threshold of 880 (corresponding to a characteristic confinement diameter of ~70 nm) and a minimum confinement duration of 10 ms to avoid spurious detections. The analysis was performed with a window size of 25 localizations and restricted to tracks at least four times longer than the window size. From this, we extracted the number of confinement events per unit time as well as the fraction of time each track spends in confined states. Following this procedure, DPPE shows a slightly higher fraction of tracks containing confinements (47%) compared to SM (39%). Among these tracks, SM exhibits more frequent confinement events, whereas DPPE spends a larger fraction of time in confined states due to longer confinement segments.

We further analysed the spatial distribution of these confinements relative to Caveolin1-positive sites by calculating radial profiles of the fraction of time tracks spend in confined states as a function of distance from the site center. This analysis indicates that DPPE exhibits more confinement overall except very close to the site center ($dr < 50$ nm), suggesting that confinement behaviour may differ locally within caveolae. Finally, we repeated the analysis separately for tracks that pass through Caveolin1-positive sites and for those that do not. This comparison did not reveal substantial differences between the two groups.

The results of this additional analysis are presented in a new Supplementary Fig. 10. Corresponding text has been added to the Results section describing the caveolae experiments (lines 274-285), and the methodology is detailed in the Methods section (lines 789-805).

R1Q4. Diffusion dynamics of sphingomyelins have been studied extensively in the literature, and there are various perturbations that affect its mobility. The authors only show a comparison between SM and DPPE, with the magnitude of the difference seemingly quite subtle compared to what has been reported by different imaging modality. As this should be a proof-of-principle study, it would be helpful for the authors to investigate SM mobility under a broader range of common perturbations used in the literature (e.g. cholesterol etc).

R1Q4R. Thank you for this suggestion. We agree that perturbation studies of sphingomyelin mobility are highly valuable for understanding lipid diffusion in cellular membranes. However, we believe that such experiments fall outside the scope of the present work.

The primary goal of this manuscript is to introduce and validate the event-triggered MINFLUX framework. The caveolae experiment, where we compare SM and DPPE diffusion at Caveolin1-positive sites and in surrounding membrane regions, is intended as a relatively simple first application. It demonstrates the event-triggered concept using static event detection (peak detection) and serves as an accessible example to introduce the technical aspects of the method. The subsequent examples of endocytic budding and virus-like particle formation build on this framework and demonstrate more advanced acquisition modes, including ROI following and time-resolved 3D MINFLUX tracking. Our intention in the lipid diffusion example is therefore not to draw broad biological conclusions, but rather to illustrate how the framework can be applied to study membrane dynamics. To clarify this point, we have revised the relevant sections of the Results to better explain the purpose of the caveolae experiments and their role as a methodological demonstration (e.g. lines 191-195 and 197-199). In addition, the expanded Discussion now more clearly outlines the advantages of the method and potential directions for future biological applications.

Nevertheless, we fully agree that systematic perturbation studies of lipid diffusion are important. In fact, such work is currently underway in our laboratory, where we are using

MINFLUX to investigate the diffusion properties of SM and DPPE under various conditions, including cholesterol depletion, and to compare the results with previous measurements using other techniques. However, these experiments address broader biological questions about membrane diffusion and do not focus on introducing the etMINFLUX framework. We therefore believe that presenting them in a dedicated study will allow the biological findings to be discussed in the appropriate depth, without being overshadowed by the methodological development that is the main focus of the present manuscript.

R1Q5. In Supplementary Fig. 1c-d, the traces seem to show irregular patterns and appear to be composed of three distinct subpopulations rather than a single one. What is the origin of this?

R1Q5R. The apparent subpopulations arise from a random scan shift occurring during individual measurements. This shift appears discretely along both scan axes, occurs unpredictably during or between frames, and remains constant until the next shift (which could be in the same frame, next frame, or later). We have not been able to identify the exact origin, but the persistence of the shift suggests that it is unlikely to arise from the scanning curve calculations in the microscope control software and is more likely related to hardware. We plan to follow up on this with the manufacturer (Abberior), as it affects not only etMINFLUX but any experiment requiring precise overlay of MINFLUX and confocal data.

The shift introduces a small, uncorrectable offset between measurements and is present in all curves shown in Supplementary Fig. 1, although it is most visible in panels c–d because the shifts happen to cluster more clearly there. To compensate for the scan shifts in general, we always estimate a spatially dependent correction value from the mean fitted curve. The majority of individual measurements lie close to this mean, although occasional shifts of up to ~50 nm can occur. Importantly, this magnitude is still well below the confocal resolution used for aligning confocal and MINFLUX data (e.g., when fitting caveolae sites) and therefore does not limit the accuracy of the downstream analysis. Due to its small amplitude, the shift is also essentially invisible in raw images. We had already mentioned this effect in Supplementary Note 3, but to clarify it further we have expanded the explanation and modified the plots so that each individual measurement is shown in a different colour, making the shifts easier to identify. Note that a single measurement can have multiple “bands”, as the shift can occur during a frame and as such for a single X line distance (fast axis), multiple shifts can be present on different Y positions (slow axis). Finally, we note that this issue can be avoided by using slightly slower confocal scanning parameters, which remain compatible with the temporal requirements of etMINFLUX. After identifying the scan shift effects, we adopted these settings, so only a subset of the caveolae datasets in the manuscript required correction.

R1Q6. Supplementary Figure 4 is difficult to interpret. Furthermore, it reports a 3D localization precision that is superior to the 2D precision, which is counterintuitive based on the specification of the commercial MINFLUX. Further clarification would be helpful.

R1Q6R. Thank you for raising this point. We have expanded the description in the Methods to clarify what is shown in Supplementary Fig. 4 and how the values were obtained.

The figure presents violin plots with individual datapoints, where each datapoint corresponds to the mean dynamic localization precision extracted as the y-axis offset of the fitted squared-displacement vs. Δt curve calculated from each individual track in a single measurement. Therefore, for the 2D datasets, each datapoint represents the mean precision obtained from the MSD fits of all tracks within one ROI. For the 3D datasets, each datapoint corresponds to one ROI cycle (i.e., repeated recordings at the same event site appear as separate datapoints). This description has now been clarified in the Methods section (lines 842-845).

Regarding the observation that the 3D localization precision appears higher than the 2D precision, we agree that this may initially seem counterintuitive given that 3D MINFLUX localization typically uses a bottle/top-hat beam rather than a donut beam. However, in practice the localization precision also depends strongly on other parameters, particularly the number of detected photons and the TCP parameter L . In our measurements, the minimum photon threshold for performing a localization was set to 15 photons for 2D tracking and 20 photons for 3D tracking, while the TCP size ($L = 100$ nm) was identical in both cases. Since MINFLUX localization precision scales with L and the photon number,² and we scale excitation power and dwell time to match our photon limit, this higher photon threshold in the 3D sequence leads to a slightly improved average precision in our datasets. These parameter choices reflect different optimization goals: the 2D tracking sequences were tuned to maximize temporal resolution by lowering dwell times and photon thresholds, whereas the 3D tracking sequences prioritize localization accuracy. To avoid confusion for readers, we have added a brief explanation of this effect in the main text (lines 475-478).

Reviewer #2 (Remarks to the Author):

The authors present the application of the event-triggered acquisition concept to MINFLUX tracking in 2D and 3D. The framework uses large confocal images to detect localized events of interest and triggers high resolution MINFLUX tracking only when and where necessary. This allows for improved throughput and the study of otherwise hard to capture biological events. Both single and multi/ROI imaging are possible using a GUI-supported open-source Python software that interacts with the commercial Imspector software via the `specpy` package and with keyboard and mouse emulation.

The performance of the framework is exemplified with three applications in the membrane diffusion space: lipid diffusion at caveolae, budding endosomes, and membrane fluidity at Gag accumulation sites. Intensity-based image analysis pipelines observe large areas and trigger localized MINFLUX tracking procedures. The authors perform diffusion analysis to compare different areas in the membrane and fit surfaces to the observed tracks in order to visualize the membrane structure.

While etMINFLUX shows promise for studying rare cellular events with unprecedented detail, better communication about the limitations of the method and its implementation would strengthen the manuscript considerably.

We thank the reviewer for the constructive comments and the overall positive assessment of our manuscript. We have carefully considered the suggestions and revised the manuscript accordingly, as detailed in the point-by-point responses below.

Major Concerns

R2Q1. The etMINFLUX concept is well introduced with clear mode descriptions. However, the connection to etSTED is largely lost in the main manuscript and not discussed deeply enough in the Supplementary Material. Looking at the Controller code, the two packages share many concepts. The authors should better explain which aspects of event-triggered imaging were

² Francisco Balzarotti et al., "Nanometer Resolution Imaging and Tracking of Fluorescent Molecules with Minimal Photon Fluxes," *Science* 355, no. 6325 (2017): 201, <https://doi.org/10.1126/science.aak9913>.

developed specifically for etMINFLUX versus adapted from etSTED. Why was the etSTED-widget-base not used with a different Inspector backend? The ROI follow modes seem like a general concept that could be used for other smart microscopy applications. Generalizing here could strengthen the manuscript's impact outside the MINFLUX space.

R2Q1R. Thank you for this helpful comment. We have expanded the discussion in the manuscript to better clarify the relationship between etMINFLUX and the previously developed etSTED framework.

In the main manuscript, we added a section in the first Results section (lines 133-139) describing the method, where we summarize the key conceptual and implementation differences between the two systems and explain why a new control widget was developed rather than directly extending the existing etSTED framework. In addition, we included a new Supplementary Note (Suppl. Note 2) that discusses these aspects in greater detail. In brief, several changes in the event-triggered control concept and the specific requirements of MINFLUX acquisitions required a substantially redesigned control architecture. As a result, the new implementation could no longer be considered a simple extension of the previously released etSTED-widget-base. Developing a standalone control widget proved to be simpler and more practical for users. The current implementation also introduces additional features, such as an optional compiled executable (mentioned on line 689 in the Methods) and JSON-based configuration files, which further motivated this redesign. We also discuss additional aspects of this comparison between etMINFLUX and etSTED in the extended Discussion section (lines 613-614, lines 617-620).

We agree with the reviewer that ROI-following modes represent a general concept that could be applied to other smart microscopy applications. We hope that the present work will encourage such developments. Indeed, we see value in eventually creating a more method-agnostic event-triggered control framework that incorporates the functionality developed here while remaining broadly adaptable across microscopy modalities. However, developing and validating such a generalized framework would be beyond the scope of the present study and would require a dedicated effort across multiple imaging methods. We have therefore added a brief comment in the Discussion outlining this perspective and potential future directions (lines 580-590).

R2Q2. For a comprehensive understanding of etMINFLUX benefits, I am missing direct quantitative comparisons of data acquisition with and without event-triggered acquisitions. How long would it have taken to acquire the data conventionally (1)? Would some results not have been possible at all, and why (2)? Where are the bottlenecks (3)? MINFLUX is especially beneficial due to low excitation intensities—how does confocal imaging influence phototoxicity in these experiments (4)?

R2Q2R. Thank you for these important questions. In response, we performed additional experiments and analyses to provide quantitative comparisons between event-triggered and manual acquisitions. These results help clarify the practical benefits and limitations of the approach. Below we address the reviewer's points individually and indicate how the manuscript has been revised accordingly.

1. Acquisition time compared to conventional experiments. We performed manual control experiments to directly compare acquisition efficiency with etMINFLUX. For the Caveolin1 experiments, we manually recorded ~10 ROIs per cell, selecting Caveolin1 peaks as ROI centers while keeping the MINFLUX acquisition parameters identical to those used with etMINFLUX. We then compared the total experiment time and defined an “overhead factor,” which quantifies the additional time spent between MINFLUX acquisitions (e.g., confocal imaging, ROI selection, and data handling) as compared to the time spent on MINFLUX

acquisition. For etMINFLUX, the average overhead factor was 1.16, corresponding to ~13% time spent on overhead. For manual acquisition, the average overhead factor was ~1.49 (~33% overhead). Thus, the non-acquisition time per ROI decreases from ~30 s (manual) to ~10 s (etMINFLUX), corresponding to roughly a threefold reduction. For Dynamin experiments, the dominant time cost is identifying the next event. In manual experiments we continuously scanned the cell confocally and initiated MINFLUX acquisitions when detecting events visually. The average time spent searching for events was ~109 s per event, compared with ~50 s per event for etMINFLUX (~2× improvement). In some cases etMINFLUX required as little as ~10 s between events, which was not achieved in manual experiments. We also compared the probability of capturing an endocytic vesicle during MINFLUX acquisition. With etMINFLUX this occurred in ~26% of cases, compared with ~9% for manual acquisition (~3× improvement). Combining this with the temporal efficiency yields roughly a sixfold increase in useful data throughput for the Dynamin experiments. These comparisons are now summarized in the Results sections (lines 244-249, lines 367-380), illustrated in Figs. 3 and 4, discussed in the first paragraph of the Discussion (lines 573-578), and described in detail in Supplementary Note 7 and new Supplementary Fig. 6.

2. Experiments that would not be feasible without event-triggering. Higher throughput directly improves the statistical power of downstream analyses. For Caveolin1 experiments, the ~3× increase in throughput improves the robustness of diffusion analyses. For Dynamin experiments, the ~6× improvement substantially increases the number of useful endocytic vesicles and number of useful vesicle-associated trajectories. Beyond numerical throughput, manual acquisition imposes additional limitations. In dynamic systems such as Dynamin events, confocal time-lapse stacks cannot easily be recorded while searching for events using the current microscope control software, forcing users to rely on continuous scanning without saving intermediate frames. This complicates post-experiment event validation and downstream analysis. Manual triggering also introduces temporal jitter and spatial inconsistencies in ROI placement. All these limitations make certain analyses such as aligning events along a common temporal axis much more difficult. For Gag experiments, manual detection proved essentially infeasible. Because the signal increases slowly over minutes, identifying the onset of events visually is extremely challenging. Even if such events would be detected, precise temporal alignment between them would be difficult. In the one case observed manually no bulging or budding occurred despite 35 minutes of following. In practice, these experiments at nascent Gag-accumulation sites were therefore only possible using automated detection in etMINFLUX. We discuss these aspects in the Results section on the Gag experiments (lines 536-544)

3. Bottlenecks. For manual experiments, the main bottleneck is identifying events reliably and rapidly enough for timely MINFLUX acquisition, especially for dynamic processes such as endocytosis or virus budding. Human reaction time, subjective detection bias, variability between cells, and manual control further limit throughput. For etMINFLUX, the primary challenge lies in optimizing event detection pipelines to achieve high recall while maintaining high precision. Detecting events early enough in the biological process is particularly important for rapid processes such as endocytosis. We discuss this trade-off in the Discussion (lines 591-608) and in Supplementary Note 6, where the performance of the optimized detection pipelines is evaluated. Future improvements may involve machine-learning-based detection approaches, which we consider a promising direction.

4. Phototoxicity considerations. MINFLUX benefits from relatively low illumination intensities compared with techniques such as STED or SMLM. In our experiments, the MINFLUX excitation intensity (~10–50 kW/cm²) corresponds to approximately 5–20 μW of equivalent

Gaussian confocal illumination³. The confocal imaging used for event detection employed 0.6–3.0 μW at 488 nm, i.e., lower power but at a shorter wavelength and therefore potentially higher phototoxicity per photon. Overall, taking also the exposure times and labelling densities of the two techniques into account, the phototoxicity contributions from confocal and MINFLUX illumination are therefore comparable under our conditions, and orders of magnitude lower than in STED or SMLM imaging experiments⁴. Importantly, etMINFLUX can reduce phototoxicity indirectly by increasing acquisition efficiency: useful datasets are obtained more quickly, reducing both MINFLUX acquisition times and the total experiment time required. Confocal imaging is also required for manual MINFLUX experiments to identify ROIs, so the event-triggered approach does not introduce an additional illumination modality. We have added a short discussion of these considerations at the end of the first paragraph of the Results section (lines 97-104), and returning to it in the Discussion (lines 578-580, lines 606-608).

The limitations for adapting to different biological applications should be better discussed. Mainly in two areas:

R2Q3. The analysis pipelines rely on intensity changes and seem similar to one another. What other approaches would be readily supported? Both by the framework and the hardware (e.g. confocal colocalization analysis, shape analysis, dynamics, neural networks).

R2Q3R. Thank you for this important point. The flexibility of the framework is indeed one of its main strengths, and we agree that this was not sufficiently emphasized in the original manuscript. We have therefore added a paragraph in the Discussion to clarify the range of possible analysis pipelines and future applications (lines 609-625). In principle, the framework is highly flexible and can support any event-detection analysis pipeline that can be implemented in Python and that operates on the information available from the confocal frames. This includes approaches based on intensity changes (as used here), but also more complex strategies such as colocalization analysis, shape-based detection, dynamic feature tracking, or classification-based approaches. In practice, the only requirement is that the analysis can be executed within a Python function and operate fast enough to provide real-time feedback for acquisition control. The framework is also compatible with more advanced computational approaches, including neural networks or other deep-learning methods, as well as GPU-accelerated analyses, provided that the control computer is equipped with appropriate hardware. For such approaches, however, the availability of sufficiently large training datasets and the inference time of the model become important practical considerations. We also note that other types of event detection previously demonstrated for etSTED can be translated to the etMINFLUX framework with only minor modifications. Overall, we believe the new etMINFLUX framework provides a general platform for implementing a wide range of event-detection strategies, which we now highlight more clearly in the revised manuscript. We further better highlight the differences between the implemented analysis pipelines with the new visual representation of the analysis pipelines in the new Suppl. Fig. 3, as described above in R1Q1R, and describe the event types we are detecting with more clarity in the end of the introduction section (lines 69-77).

³ Klaus C. Gwosch et al., “MINFLUX Nanoscopy Delivers 3D Multicolor Nanometer Resolution in Cells,” *Nature Methods* 17, no. 2 (2020): 2, <https://doi.org/10.1038/s41592-019-0688-0>; Lukas Scheiderer et al., “MINFLUX Achieves Molecular Resolution with Minimal Photons,” *Nature Photonics* 19, no. 3 (2025): 238–47, <https://doi.org/10.1038/s41566-025-01625-0>.

⁴ Jonatan Alvelid and Ilaria Testa, “Fluorescence Microscopy at the Molecular Scale,” *Current Opinion in Biomedical Engineering* 12 (December 2019): 34–42, <https://doi.org/10.1016/j.cobme.2019.09.009>.

R2Q4. The presented applications are limited to membrane diffusion. What are the limitations to applying this to other biological problems e.g. motor tracking in 3D?

R2Q4R. Thank you for raising this point. We would like to emphasize that there is nothing inherent in the framework that restricts etMINFLUX to membrane diffusion studies. Rather, the main limitations arise from those of MINFLUX itself, particularly the labelling strategies and fluorophore densities required for reliable single-molecule tracking. In the examples presented here, lipid analogues were used as probes, enabling both membrane diffusion analysis and topographical studies. However, the framework is not limited to such applications. In principle, it can be applied to a wide range of targets, including membrane proteins, intracellular proteins, or other labelled molecules in subcellular compartments or transport pathways. Instead, we believe that the main practical constraint lies in the type of biological processes that benefit most from event-triggered acquisition. The approach is particularly advantageous for processes with relatively rapid onset and characteristic timescales on the order of tens of seconds, such as the endocytic events studied here, where automated detection can significantly increase throughput and enable systematic acquisition at the relevant time points; or for slow processes that are difficult to manually detect, such as nascent virus budding sites. To clarify this perspective, we have added a paragraph in the Discussion (lines 645-657) outlining the types of biological questions where we expect etMINFLUX to be most beneficial, as well as its broader potential applications. As an example of ongoing work beyond membrane diffusion, we are currently applying the framework to track peroxisomal membrane proteins using a two-color confocal event detection pipeline that identifies temporally stationary peroxisomes, which otherwise are very mobile on short timescales, with appropriate labelling density for MINFLUX tracking. Manual acquisition in this case severely limits data throughput due to the difficulty of manually assessing the two-color data, selecting such peroxisomes, and initiating a MINFLUX acquisition before the situation has changed. Please see an example of such an experiment in Rebuttal Figure 1.

Rebuttal Figure 1. EtMINFLUX applied to peroxisomal membrane protein tracking. **a.** Sketch of the biological system (left), with the peroxisome (sphere), membrane proteins (black outline), dye (red), and MINFLUX trajectory (dashed red line). The ideal, low-density labelling situation is shown, with one to a few proteins labelled on the peroxisome. An example overlay

between confocal image (fire) and MINFLUX trajectories (gray lines) from an etMINFLUX measurement (right). **b.** Temporally color-coded confocal image stack, showing the movement of peroxisomes. The peroxisome of the detected event had a mean frame-to-frame movement smaller than a set threshold. **c.** Two-color confocal image, with peroxisomal lumen protein (magenta) and low-density peroxisomal membrane protein as labelled for MINFLUX tracking (green). The peroxisome (peak detected in lumen signal) of the detected event had a membrane protein signal level above a minimum threshold, to ensure at least one labelled membrane protein for MINFLUX tracking, but below a maximum threshold, to ensure not too high membrane protein labelling density. **d.** Confocal and MINFLUX data from the detected event. Confocal image before (left) and after (middle) MINFLUX acquisition, and confocal and MINFLUX data overlaid (right). **e.** MINFLUX data from the detected event site, shown projected in XY (left) and in a 3D view (right), with the z coordinate color-coded. **f.** Cross-section of the MINFLUX data, along the equatorial plane of the peroxisome, indicating the hollow spherical shell that the membrane protein trajectories are forming when moving on the peroxisomal membrane. **g.** MINFLUX data from the detected event site, shown projected in XY (left) and in a 3D view (right), with the time color-coded, showing MINFLUX localizations from the start (0 s) to the end of the MINFLUX acquisition (last localization at 1.6 s).

R2Q5. As a reader from the technical side, I have difficulty understanding the impact of the biological findings. Some diffusion detail could be shifted to Supplementary Material, allowing the main manuscript to provide higher-level biological context with explanations for technical readers regarding the data's impact.

R2Q5R. Thank you for this comment. As noted in our previous responses (e.g., R1Q4R), the primary aim of this manuscript is to present the methodological and technical advances of the etMINFLUX framework. We believe that the revisions made throughout the manuscript now highlight these aspects more clearly. At the same time, we agree that it is helpful to place the biological test systems used here into a broader biological context. To address this, we have expanded the opening paragraphs of each Results subsection (lines 204-207, lines 325-330, lines 454-458) describing the experimental applications, providing additional context on the biological processes studied and explaining why the extracted parameters are relevant for future investigations. In addition, the Discussion section has been substantially expanded to better position the work within the broader context of potential biological applications and future developments.

The analyses presented for the three example systems are intended to illustrate the types of parameters that can be extracted using this framework rather than to provide exhaustive biological conclusions. As mentioned above (see also R1NR1 and R3Q1R), we are currently pursuing separate studies that use MINFLUX to investigate lipid diffusion of SM, DPPE, and related lipids in greater depth, as well as studies applying etMINFLUX to virus budding sites and other biological processes. We believe that presenting those biological investigations in dedicated manuscripts will allow the biological findings to be discussed in the appropriate depth, whereas including them here would shift the focus away from the methodological contributions of the present work.

Regarding the level of analysis detail presented in the main text, we have shortened the paragraphs on diffusion analysis at caveolae sites by removing information about the various analyses that are already present in respective Methods sections and/or Supplementary Notes as well as clarified the language to make the results clearer. We believe that the set of diffusion analysis presented in the main text and main figures are of importance to describe the possibilities that the throughput of etMINFLUX opens in this and similar applications.

Minor Concerns

R2Q6. The authors state in the discussion that application to MINFLUX imaging would be straightforward. While I agree technically, applications of smart microscopy for fixed samples are typically harder to identify and dynamic imaging is hard using MINFLUX. I also see the combined complexity of MINFLUX imaging plus an event-triggered framework as being greater than for tracking, making this adaptation not as trivial as it might seem at first.

R2Q6R. We agree with the reviewer that smart microscopy applications for fixed samples are generally less common than for dynamic imaging. However, we believe that etMINFLUX could still be beneficial for certain MINFLUX imaging applications where increased throughput and reduced manual intervention are advantageous. For example, automated detection and acquisition of multiple similar subcellular structures such as nuclear pore complexes within a cell or across neighboring cells could be implemented by triggering ROIs based on peak detection. In combination with labelling approaches such as DNA-PAINT, and with the possibility to control activation lasers through the etMINFLUX framework as now implemented, such experiments could in principle be automated effectively. To better reflect these considerations, we have revised the relevant paragraph in the Discussion to clarify and exemplify both the potential opportunities and the practical limitations of applying event-triggered approaches to MINFLUX imaging (specifically lines 587-590).

R2Q7. I can't make much sense of the denser diffusion plots (e.g., Fig. 4f)

R2Q7R. Thank you for pointing this out. To improve readability, we have updated all dense trajectory plots (Figs. 4f, 5g, 5h; Extended Data Figs. 4a–d (also increased dpi), 5a–d (also increased dpi), 6b, 6d, 8a–d; figures with already low density such as Extended Data Figs. 6f and 7b were left unchanged) to display only a subset of tracks, similar to the approach used in Fig. 3d. Specifically, tracks are first ordered temporally and then subsampled by selecting every N -th track, where N is chosen to retain a defined percentage of tracks in the final plot. This produces a representative subset that remains evenly distributed across the full recording period. The percentage shown was selected individually for each dataset and typically corresponds to ~30% of tracks (range ~30–60%). The exact fraction displayed is indicated in each figure and legend. We also slightly reduced the line thickness to improve the visual separation of individual trajectories and localizations. For completeness, we added a new Supplementary Fig. 15 showing the original plots with 100% of trajectories/localizations in an enlarged format for datasets where subsampling was applied. We hope these changes make the figures clearer to interpret.

R2Q8. It would be great to have compiled dynamic data as videos to get a better feeling of the dynamics of the experiments. Or/and even a napari plugin or script to load the confocal data together with the tracks.

R2Q8R. Thank you for these suggestions, which we have both implemented. Firstly, we have developed a *napari* plugin (github.com/jonatanalvelid/napari-etminflux-data-viewer and available from PyPI: <https://pypi.org/project/napari-etminflux-data-viewer/>) that allows users to load and explore etMINFLUX datasets interactively. The plugin supports all acquisition modes presented in the manuscript and both 2D and 3D tracking data. Confocal and MINFLUX datasets are loaded into a common coordinate system, allowing them to be viewed separately or overlaid. MINFLUX localizations are displayed according to their absolute acquisition time, enabling concatenation of multiple acquisition cycles, while confocal time-lapse frames are updated according to their recording time. Users can scroll through the temporal dimension using an interactive slider that synchronously updates both datasets, as well as add a real-time time stamp label showing the real experiment time centered on the event detection ($t=0$). Additional details on functionality and installation are available on the GitHub repository. We

now mention the *napari* data viewer in the Results section (lines 186-190), in the Methods section (lines 833-834), in the Code availability section (lines 870-873) and describe it in more detail in a new Supplementary Note 4. In addition, the scripts used for post-acquisition analysis and visualization of the datasets presented in this work are publicly available (<https://github.com/ionatanavelid/etMINFLUX-analysis-public>), as stated in the Code Availability section (lines 867-870). To clarify this point, we have added a short Methods subsection on Post-acquisition data analysis (lines 768-773).

Secondly, we have added supplementary movies showcasing etMINFLUX datasets for each experiment (Suppl. Movies 1–8; 2× Caveolin1 experiments, 3× Dynamin1 experiments, 3× Gag experiments). In the movies, the confocal timelapse and MINFLUX localizations are appearing in real time, and MINFLUX acquisition cycles are concatenated after each other. In data from non-following MultiROI recording modes the MINFLUX datasets all start from $t=0$, despite being recorded subsequently. The movies were generated using the *napari-animation* plugin together with the newly developed *napari-etminflux-data-viewer*, as mentioned in the Methods section (lines 833-834).

R2Q9. The performance values for the analysis pipelines are relatively low. The authors should discuss the impact and type of false positives and how the performance of these pipelines could be further improved.

R2Q9R. We have expanded the discussion of pipeline performance in the Results sections for the Dynamin1 (lines 397-407) and Gag (lines 512-524) experiments, as well as in the substantially extended Discussion (lines 592-609). Together with the newly added offline optimization of pipeline parameters (see R1Q1R, point 4), this now provides a more thorough evaluation of the detection pipelines and their achievable performance. In brief, while the precision values in the real-time experiments are moderate, the offline optimization demonstrates that the Dynamin1 accumulation pipeline can reach a precision of up to 1.0, albeit at the expense of reduced recall. In practice, however, maintaining such optimal performance is challenging because variations in sample preparation, labelling density, and imaging conditions affect the relevant thresholds. Importantly, the framework records all relevant confocal and MINFLUX data together with metadata, allowing detected events to be inspected and filtered during post-acquisition analysis. For the Gag accumulation pipeline, the Results section already discussed the most common source of false positives, and we have now expanded this discussion with the results from the offline parameter optimization. Due to the relatively small number of events present in such experiments and datasets, optimization, whether manual or automated, is inherently difficult, and the current pipeline likely operates close to its practical performance limits under these conditions. As with the Dynamin1 analysis, more sophisticated approaches, including machine-learning-based detection, could further improve both precision and recall at the expense of complexity and inference times. These trade-offs and potential future improvements are now discussed more extensively in the Discussion section. In particular, we note that implementing machine-learning-based event detection is a direction we are actively considering for future studies using the etMINFLUX framework.

Reviewer #2 (Remarks on code availability):

Control

R2Q10. The repository is well documented and the code is well structured. I was not able to test the software, as newer specpy versions are only available with lmspector. I tried to get the simulation environment working under Python 3.6 with the publicly available specpy 1.2.1,

which was not possible due to several compatibility issues. I think the authors should comment more extensively on how they view this situation, especially given that the public *specpy* version was last updated in 2017. Is there a commitment from Abberior to maintain compatibility in the future? Are there alternative ways to run the software if newer or updated systems no longer support *specpy*?

R2Q10R. Thank you for this thoughtful comment. While the etMINFLUX concept itself is independent of a specific implementation, the current software implementation does rely on the *specpy* interface provided with abberior's Imspector control software. As *specpy* is under continuous development by abberior and distributed together with each Imspector release, users operating an abberior MINFLUX system will always have access to the compatible version required to run the framework. Based on our communication and experience, abberior continues to maintain and extend this interface. For example, in the most recent Imspector release (m2410), support for setting MINFLUX ROIs directly through *specpy* has been introduced. This removes the need for simulated mouse and keyboard input used in the earlier version of etMINFLUX and significantly improves the robustness of the implementation. The repository has been updated accordingly: the current main branch supports the newer Imspector version, while the earlier implementation for version m2205 remains available in a separate branch for users who have not yet updated their system. More generally, this type of dependency is common for software interacting with commercial microscope control systems. If future control software versions were to replace *specpy* with a different API, the framework would need to be adapted accordingly, but the modular design of the codebase should make such transitions feasible.

Regarding the availability of *specpy*, while newer versions are not distributed publicly outside the Imspector environment, any user operating an abberior MINFLUX system will have access to the appropriate version. To facilitate testing and exploration for users without direct access to such hardware, we have now implemented a fully functional simulation mode that allows the analysis pipelines to be tested and optimized in an etMINFLUX environment (see response R2Q13R).

To clarify these aspects, we added a short note on the required *specpy* version in Supplementary Note 2 and expanded the documentation in the repository README.

R2Q11. The use of emulated mouse control seems fragile to me. The authors mention in the Supplementary Material that an extension of the API would solve this issue. Are there efforts underway to implement such an extension? Otherwise, I assume the system cannot be touched during the experiment to avoid interfering with the simulated mouse control. In this case, the authors should temper the parts in the manuscript that mention on-the-fly adjustments to imaging parameters.

R2Q11R. As mentioned in our response R2Q10R, the most recent Imspector release (m2410) includes significant extensions to the *specpy* API, allowing MINFLUX ROIs to be defined, measurements to be started, and repeated acquisitions to be controlled directly through the API. The updated version of etMINFLUX (available in the main branch of the repository) now uses this functionality and no longer relies on simulated mouse or keyboard input. As a result, the system can run without requiring Imspector to remain the active window. This also allows normal user interaction with the workstation during experiments, for example inspecting data or adjusting compatible parameters in the etMINFLUX interface while acquisitions are running. For completeness, the previous implementation designed for Imspector version m2205 remains available in the repository (branch *m2205*) for users who have not yet updated their software. In that earlier version, emulated mouse input was required because the necessary API functions were not available. Even there, the emulation was limited to a few specific actions, primarily selecting the MINFLUX ROI and starting or stopping confocal imaging.

During ongoing MINFLUX acquisitions or when confocal monitoring runs at slower frame rates, on-the-fly adjustments were still possible with appropriate care.

To clarify these points, we updated the relevant descriptions in the main Results (removal lines 139-141) and Methods (lines 690-693) sections, Supplementary Note 1, and the repository README to reflect the improvements introduced with the newer *Imspector/specpy* version and therefore *etMINFLUX* version and to explain the differences between the two implementations.

R2Q12. While the documentation of the setup and calibration steps is extensive and understandable, I see several problematic areas that need improvement to ensure reusability. The mention of specific line numbers in the code is prone to becoming outdated with code updates. I think these settings should be extracted to external configuration files with clearly structured and named fields. The calibration procedure for pixels might not work for zoomed views, and new users will need considerable expertise to optimize timing settings. A more streamlined process would ensure better reusability by testing timing on a specific system and suggesting optimized values for example.

R2Q12R. We thank the reviewer for carefully examining the codebase and for these constructive suggestions, which have helped us improve the usability of the software.

External configuration files. We have implemented support for external configuration files to manage all user-defined and microscope-specific settings. These parameters are now stored in a structured `.json` file that is loaded at startup (a default file is generated upon first startup in an *etMINFLUX* folder in the user's Documents folder and can then be adjusted). Several settings previously exposed in the GUI have also been moved to this configuration file to simplify the interface. As a result, references to specific line numbers in the code have been removed from the documentation, and users no longer need to modify the source code directly. The configuration fields are designed to be self-explanatory and are described in detail in the repository README and experiment guide. Thanks to these updates, we have also been able to release an executable of the latest version (*m2410*) which only requires the addition of *specpy* in the folder of the executable to run (unless simulation mode is used, see response R2Q13R), not included in the executable for proprietary reasons.

Pixel calibration. We agree that the pixel calibration procedure in the older implementation is somewhat cumbersome and sensitive to zoomed confocal views, as the reviewer noted. This limitation arises because ROI selection in *Imspector* version *m2205* is controlled entirely via mouse input, without accessible parameter fields in the API. While not ideal, this was the only practical way to enable *etMINFLUX* experiments in that environment. In the newer *Imspector* release (*m2410*), the necessary control functions have been added to the *specpy* interface, allowing ROI handling and related parameters to be set directly through the API (see responses R2Q10R and R2Q11R). The updated *etMINFLUX* implementation therefore no longer requires this calibration procedure and is significantly more streamlined. We updated the text, particularly Supplementary Note 1, to reflect the improvements and clarify the improved workflow, and now additionally explain the differences between the two versions.

Timing settings. The default timing parameters provided with the software were selected as conservative values that should function reliably across different systems. While they may introduce small overheads on some setups, these delays are not the dominant time-limiting factors in the experiments. For this reason we did not implement an automated timing calibration routine at this stage. To reduce confusion for new users, these parameters have been moved from the GUI into the configuration `.json` file. Advanced users can still adjust them if needing temporal optimization for their system. The repository README now includes additional explanations of the purpose of these timing parameters and guidance on how they can be optimized. Furthermore, the updates in the new *Imspector* version, as previously

described, removes the emulated mouse and keyboard control, and as such some of the delays required previously have been removed.

R2Q13. For users without direct access to a MINFLUX system (for example, when using it at a core facility), it would be important to have a simulation environment to test analysis workflows in the context of etMINFLUX on existing data. Given the problematic dependencies, this seems non-trivial, even with the mocked *Imspector* instance.

R2Q13R. Thank you for this excellent suggestion. We agree that providing a simulation environment is important for users who do not have direct access to a MINFLUX system, for example when preparing experiments or testing analysis pipelines on existing datasets. While a mocked *Imspector* instance was initially implemented, we have now added a simpler and fully functional simulation mode to address this need. The simulation functionality is implemented through a separate set of Controller and Widget classes. At startup, the software checks whether the user requested simulation mode in the `setup.json` configuration file. It then attempts to load *specpy* (either from the folder of the script/executable or installed in the environment that runs the script), and establish a connection to *Imspector*. If simulation mode is requested, or if *specpy* cannot be loaded or *Imspector* cannot be reached, the software automatically launches in simulation mode. In this mode, hardware-specific functionality is disabled or greyed out to simplify the interface while keeping the overall GUI structure similar to the real environment. Analysis pipelines can still be loaded as usual, and users can test and optimize them using previously recorded confocal time-lapse data and running test experiments. To support this workflow, a field has been added to load a confocal TIFF stack, which is then processed frame by frame during a simulated experiment. Users can also define the update rate at which frames are read and analysed, allowing realistic testing of event detection pipelines. This design allows the simulation mode to operate entirely without *specpy* or an active *Imspector* connection. The software can therefore be run either through the compiled executable or within a Python environment installed using the provided `.yml` file. We have added a brief description of the simulation mode in the Results section of the manuscript (lines 148-150), mention its benefits in the Discussion section (lines 610-611), and provide further details in Supplementary Note 1 and in the repository README.

Analysis

The analysis code is structured well and data flow is well documented.

Reviewer #3 (Remarks to the Author):

The manuscript "Event-triggered MINFLUX microscopy: smart microscopy to catch and follow rare events" by Alvelid et al. addresses a highly relevant and timely issue namely how to bridge the very limited spatial throughput of the new and powerful microscopy modality MINFLUX with imaging in living cells where everything moves and relevant things can happen and appear at unknown positions in both space and time. To resolve this conundrum the authors have developed an event-triggered imaging approach termed etMINFLUX. With this approach the authors either image a single large field-of-view (FOV) and algorithmically identify multiple sites of interest and do targeted MINFLUX imaging at these sites. Alternatively, the authors let the automated control software monitor a large FOV at regular intervals with standard confocal microscopy and subsequently upon identification of an "event" corresponding to a predefined criteria do targeted MINFLUX imaging within a small FOV around the identified event. The

authors apply the method to three different biological systems: 1, the diffusion of lipids through caveolae; 2, endosomal budding from the plasma membrane; and 3, lipid diffusion and shape determination of Gag-protein positive HIV-1 (human immunodeficiency virus) budding sites. The work is clearly presented and highly relevant. Indeed, from my perspective, the presented approach seems to be basically a necessary add-on to most real-world scientific questions one would want to address with the high resolution MINFLUX technique given this technique's limited FOV. The obtained results speak to the usefulness of the approach as it would have been difficult to obtain such large numbers of observations as reported (of the three different examples) without the use of the automated strategy presented. There are a few limitations and minor issues presented below but in general the validity and usefulness of the approach is very sound, both from a theoretical and a practically achieved perspective.

We thank the reviewer for the thoughtful assessment of our work and for the positive evaluation of the approach and its relevance. We appreciate the constructive feedback and have addressed all comments and suggestions in the revised manuscript, as detailed in the responses below.

Minor issues:

R3Q1. In figure 3 the magnitude of the differences between the diffusion properties of both the two studied lipids (SM and DPPE) and within and outside of caveolae appear quite small. This is not a critique of the usefulness of the method or the validity of the results but as an example of effects on lipid diffusivity it is perhaps not very striking, which to some extent limits the interpretation of what sensitivity/dynamic range of measurable diffusion changes within structures can be measured with this approach.

R3Q1R. Thank you for this comment. We would like to emphasize that the sensitivity to diffusion changes in etMINFLUX is fundamentally the same as in conventional MINFLUX. The main advantage of the event-triggered framework is the increased throughput, which allows the collection of larger datasets and thus improves statistical power when analysing subtle diffusion differences. However, the intrinsic measurement sensitivity itself is not altered. For studies exploring the limits of MINFLUX for diffusion measurements and its application to characterize membrane dynamics in more depth, we refer to recent work from our laboratory (Vogler et al., 2025⁵; Reina et al., 2025⁶). More generally, the ultimate limits of MINFLUX for diffusion studies are still being explored, and we are continuing work in this direction, for example comparing lipid diffusion measured with MINFLUX to results obtained with other techniques, as also mentioned above (R1Q4R). In the present manuscript, the caveolae experiment was intentionally chosen as a relatively simple first application to demonstrate the event-triggered approach in a 2D system. The subsequent examples increase in complexity, moving toward 3D tracking and more dynamic processes where the advantages of etMINFLUX become more pronounced. Our goal was therefore not to present a biologically striking diffusion effect, but rather to demonstrate how the framework enables systematic data acquisition and quantitative analysis in such systems. In the end, the small differences observed might not have been possible to see in a more data-limited manual-acquisition study,

⁵ Bela T. L. Vogler et al., "Parameter Optimization for MINFLUX Microscopy Enabled Single Particle Tracking," *Communications Biology* 8, no. 1 (2025): 1573, <https://doi.org/10.1038/s42003-025-09060-1>.

⁶ Francesco Reina et al., "Concurrent Diffusion of Nicotinic Acetylcholine Receptors and Fluorescent Cholesterol Disclosed by Two-Colour Sub-Millisecond MINFLUX-Based Single-Molecule Tracking," *Nature Communications* 16, no. 1 (2025): 6336, <https://doi.org/10.1038/s41467-025-61489-4>.

therefore also indicating one of the strong benefits of etMINFLUX. In response to this and related reviewer comments, we have revised the manuscript to clarify the role of these experiments as methodological demonstrations throughout the full text. Specific clarifications were added at the end of the Results section on the method (lines 191-195), and in the introductory paragraphs of the Results sections describing the three applications (line 197-199, lines 318-319, line 449), and throughout the Discussion.

R3Q2. In figure 4f and g, example membrane topology maps and their relative frequency at dynamin1 sites are presented. To get a better appreciation of the specificity of the endosomal invaginations it would have been valuable to see a similar analysis done on random non-dynamin1 positive sites.

R3Q2R. Thank you for this helpful suggestion. We agree that including such a control strengthens the interpretation of the Dynamin1-triggered experiments. To address this, we performed additional etMINFLUX measurements using randomized event sites. These were selected from a binary mask of the cell surface that excluded Dynamin1-positive regions, resulting in 42 Dynamin1-negative-centered sites. At these positions, we carried out single-ROI-following etMINFLUX experiments over 4-6 acquisition cycles of 30 s each, using the same confocal and MINFLUX acquisition parameters as in the original experiments. All 42 sites remained Dynamin1-negative throughout the recordings. Among these, 4 sites (~9.5%) showed a membrane-bound cytosol-facing vesicle at some point during the observation period; however, none of these vesicles overlapped with Dynamin1 signal. In contrast, in the Dynamin1-triggered experiments, 39% of the 100 confirmed Dynamin1-positive sites showed vesicle-like membrane invaginations within the ROI, while 26% of the sites showed vesicle-like invaginations overlapping with Dynamin1 signal. These results demonstrate the specificity of the event-triggered detection and support the interpretation that the vesicles observed in the Dynamin1-triggered experiments correspond to endocytic events. The analysis is now presented in a new Supplementary Fig. 12, which includes detection statistics, an example of a Dynamin1-negative event shown analogously to Fig. 4, and comparison matrices showing confocal–MINFLUX overlays and 3D reconstructions for Dynamin1-positive and randomized Dynamin1-negative sites. The Results section describing the dynamin1 experiments has been updated accordingly with the addition of a paragraph (lines 418-426).

R3Q3. Similar to issue #2 above, in figure 5g-l it would have been very informative to see a similar analysis of random membrane sites.

R3Q3R. Thank you again for this suggestion, which we agree is equally relevant for the virus budding experiments. As a control analogous to the analysis described in R3Q2R, we performed additional multi-ROI-following experiments at randomized Gag-negative membrane sites. These sites were selected using the etMINFLUX widget from a binary mask of the cell surface excluding Gag-positive regions, and the experiments were conducted using the same confocal and MINFLUX acquisition settings as in the original measurements. In total, 41 Gag-negative-centered sites were recorded, each followed for 4–16 acquisition cycles of 60 s (corresponding to ~12–48 minutes of observation time). All 41 sites remained Gag-negative throughout the recordings. None showed consistent or localized outward membrane bending characteristic of virus budding. While the membrane topology varied locally, as also observed in the Gag-positive ROIs, no persistent bulging or budding structures were detected in any of the randomized Gag-negative sites. These results further demonstrate the specificity of the event-triggered detection and support the interpretation that the membrane deformations observed in the Gag-triggered experiments correspond to bona fide HIV budding sites. The results are presented in a new Supplementary Fig. 14, which includes detection statistics, an example of a randomized control site displayed in the same format as Fig. 5, and comparison matrices showing confocal–MINFLUX overlays and 3D reconstructions for Gag-positive and

randomized Gag-negative sites. The Results section describing the Gag experiments has been updated accordingly with the description of these experiments (lines 529-537).

Reviewer #3 (Remarks on code availability):

The code and documentation appear well organized and possible to follow to install and use. I have not reviewed the code in detail as this is beyond my expertise.

We thank the reviewer for evaluating the code and documentation from a user perspective, which is equally valuable. To further lower the barrier for non-expert users, we have now added a compiled executable for running the etMINFLUX widget, as described above (R2Q1R), which allows the software to be used without setting up a full Python environment.